# Plasma-to-tumour tissue integrated proteomics using nano-omics for biomarker discovery in glioblastoma

Xinming Liu [1,2,3], Hanan Abmanhal-Masarweh[1,3], Olivia Iwanowytsch[1,3], Emmanuel Okwelogu[1,3], Kiana Arashvand[2,3], Konstantina Karabatsou[4], Pietro Ivo D'Urso[4], Federico Roncaroli [5,6], Kostas Kostarelos [3,6,7,8,9], Thomas Kisby [2,3,6] ✉ & Marilena Hadjidemetriou [1,3,6] ✉

Glioblastoma (GB) is the most lethal brain cancer, with patient survival rates remaining largely unchanged over the past two decades. Here, we introduce the Nano-omics integrative workflow that links systemic (plasma) and localised (tumour tissue) protein changes associated with GB progression. Mass spectrometry analysis of the nanoparticle biomolecule corona in GL261-bearing mice at different stages of GB revealed plasma protein alterations, even at low tumour burden, with over 30% overlap between GB-specific plasma and tumour tissue proteomes. Analysis of matched plasma and surgically resected tumour samples from high-grade glioma patients demonstrates the clinical applicability of the Nano-omics pipeline. Cross-species correlation identified 48 potential GB biomarker candidates involved in actin cytoskeleton organisation, focal adhesion, platelet activation, leukocyte migration, amino acid biosynthesis, carbon metabolism, and phagosome pathways. The Nano-omics approach holds promise for the discovery of early detection and disease monitoring biomarkers of central nervous system conditions, paving the way for subsequent clinical validation.

Glioblastoma (GB) is the most common and aggressive primary tumour of the central nervous system (CNS) in adults, with an incidence of 3 to 6 cases per 100,000 adults annually[1–4]. Despite the considerable progress in surgical techniques, radiotherapy, and chemotherapy modalities, the prognosis for patients remains poor, with a 5-year overall survival rate of less than 5% and a median overall survival of 12–18 months[1,5].

Such dismal outcomes have fuelled efforts to understand the molecular landscape of GB, with the goal of achieving more precise patient stratification for personalised treatment[3]. The identification of peripheral biomarkers in 'liquid biopsies' has the potential to capture the spatial and temporal heterogeneity of GB, contributing to improved diagnosis, treatment response monitoring, and early detection of recurrence[6]. However, despite recent advances in multi-

[1]NanoOmics Lab, Division of Cancer Sciences, School of Medical Sciences, Faculty of Biology, Medicine and Health, The University of Manchester, Manchester, UK. [2]NanoTherapeutics Lab, Division of Cell Matrix Biology and Regenerative Medicine, School of Biological Sciences, Faculty of Biology, Medicine and Health, University of Manchester, Manchester, UK. [3]Centre for Nanotechnology in Medicine, Faculty of Biology, Medicine and Health, The University of Manchester, Manchester, UK. [4]Department of Neurosurgery, Manchester Centre for Clinical Neurosciences, Salford Royal NHS Foundation Trust, Manchester, UK. [5]Division of Neuroscience, School of Biological Sciences, Faculty of Biology, Medicine and Health, The University of Manchester, Manchester, UK. [6]Geoffrey Jefferson Brain Research Centre, Manchester Academic Health Science Centre, Northern Care Alliance NHS Foundation Trust, The University of Manchester, Manchester, UK. [7]Nanomedicine Lab, Catalan Institute of Nanoscience and Nanotechnology (ICN2), CSIC and BIST, Campus UAB, Barcelona, Spain. [8]Institute of Neuroscience, Universitat Autònoma de Barcelona, Barcelona, Spain. [9]Institució Catalana de Recerca i Estudis Avançats (ICREA), Pg. Lluís Companys, Barcelona, Spain. ✉e-mail: thomas.kisby@manchester.ac.uk; marilena.hadjidemetriou@manchester.ac.uk

tumour early detection tests, the clinical application of liquid-biopsies in brain tumours is not fully explored, with no clinically approved biomarkers currently available for detection or disease monitoring.

A considerable challenge lies in deciphering the correlation between tumour pathophysiology with blood-borne proteomic alterations. Such challenge is largely due to the overwhelming 'masking effect' of highly abundant proteins, such as albumin[7]. The extraction and proteomic analysis of considerably diluted amounts of brain disease-specific proteins in blood, pose a significant bottleneck for GB biomarker discovery. This underscores the need for the development of novel analytical tools to enable the discovery of brain disease-induced alterations in the blood proteome.

To address this limitation, our previous work demonstrated the potential of exploiting the nanoparticle (NP) biomolecule corona to overcome the albumin masking effect and enrich low-abundance proteins[8,9]. By leveraging the spontaneous surface-absorption of biomolecules onto the NP surface in vivo (following intravenous administration of NPs) and ex vivo (through incubation of NPs with plasma samples),[10] the 'Nano-omics approach' has been applied for biomarker discovery in both preclinical and clinical settings of cancer and neurodegenerative diseases[11–16].

The potential of multidimensional proteomics in GB biomarker discovery was recently exemplified in a human cohort study integrating tumour tissue and peripheral blood proteomics[17]. However, most clinical studies analyse samples from single disease stages, limiting our understanding of longitudinal proteomic changes. Moreover, in the context of GB, obtaining human clinical samples at low tumour burden remains extremely challenging.

Here, we build upon our previously established NP-enabled plasma proteomics workflow by integrating spatially resolved tumour tissue proteomics analysis. Our goal is to identify highly specific plasma proteomic signatures associated with GB while elucidating the underlying molecular mechanisms that drive the systemic response to GB growth. By bridging the gap between plasma and tumour tissue proteomes, we adopt a comprehensive, mechanism-driven approach to biomarker discovery, linking systemic proteomic alterations with those observed in the GB tumour microenvironment. We utilise the GL261 syngeneic murine model to longitudinally track the plasma and tumour tissue proteomes at various stages of GB progression.

Our findings reveal a notable overlap between the NP-enriched plasma and tumour tissue proteomes, even at low tumour burden, highlighting dynamic protein changes associated with GB progression. In addition, we provide pilot data demonstrating the applicability of the Nano-omics 'plasma-to-tumour' integrative pipeline in clinical biomarker discovery by analysing matched plasma and surgically resected tumour samples from $n = 10$ patients with high-grade gliomas. Finally, our proof-of-concept cross-species pathway correlation analysis facilitated the selection of a panel of 48 protein biomarker candidates for subsequent clinical validation. The Nano-omics integrative approach has the potential to provide valuable biological insights into a range of conditions affecting the central nervous system, paving the way for mechanism-driven biomarker discovery and personalised treatment strategies.

## Results and discussion
### Nanoparticle-enabled plasma proteomics in the GL261 mouse model of GB
To allow longitudinal analysis of the plasma proteome at different stages of GB progression, we employed the orthotopic GL261 murine model. This syngeneic model offers highly reproducible tumour growth kinetics in immunocompetent mice[18,19]. Despite its limitations, the GL261 model recapitulates several features of human GB, including high cell proliferation, neo-angiogenesis and infiltration of the adjacent brain tissue[19–21].

The GL261 cells were intracranially injected into the striatum of C57BL/6 J mice, whilst control mice received a vehicle injection of the same volume. Matched blood and tumour tissue samples were collected at day 7 (D7), day 14 (D14) and day 18 (D18) post-tumour inoculation, representing low, medium and large tumour volumes, respectively (Fig. 1a). Histopathological examination of the tumour was performed on brain tissue sections stained with haematoxylin and eosin (H&E) and revealed progressive tumour growth at the three different time points (Fig. 1b, c). As illustrated in Fig. 1b, newly formed vessels were apparent at D14 and became prominent by D18, alongside variable cellularity indicative of a heterogenous microenvironment, focal necrosis and infiltrative growth.

To enable longitudinal in-depth analysis of the plasma proteome, we molecularly characterised the in vivo protein corona formed onto intravenously injected liposomes (Fig. 1a and Supplementary Fig. 1a) at D7, D14 and D18 post-tumour inoculation. Based on our previous in vivo work using the same clinically used liposome formulation (HSPC: Chol: DSPE-PEG2000), a molecularly complex protein corona is formed as early as 10 min post-injection[8,22], while the surface functionalisation of liposomes with polyethylene glycol (PEG) ensures colloidal stability of the corona-coated liposomes and long blood circulation half-life[9,23]. Blood-circulating liposomes were recovered by cardiac puncture 10 min post-intravenous administration of liposomes in GL261-bearing and saline-injected control mice. Corona-coated liposomes were subsequently recovered and purified by size exclusion chromatography and membrane ultrafiltration (Fig. 2a). As we have previously described[8,9,22,24,25], this two-step protocol effectively removes unbound plasma molecules and vesicles while allowing for the characterisation of proteins attached to the NP surface, either individually or in complex with other proteins. While the characterisation of proteins within liposome-associated extracellular vesicles is possible, it warrants further investigation.

Our data showed increased nanoparticle protein binding (Pb) values with progressive increase of the tumour volume at D7, D14 and D18 (Fig. 2b). In addition, the Pb values were significantly higher between the GL261-bearing and control mice at D14 and D18, suggesting the formation of a molecularly enriched protein corona (Fig. 2b). This finding is consistent with our previously published work investigating protein corona formation in melanoma and lung adenocarcinoma-bearing mice[12]. Control experiments quantifying the total protein amount in plasma samples obtained from tumour-bearing mice revealed no significant differences at D7, D14, and D18 (Supplementary Fig. 1b). We, therefore, attribute the elevated Pb values to the increased tumour burden, which enhances the release of tumour-secreted materials into the bloodstream, increasing their availability for NP binding. While quantitative assessment of Pb values could potentially indicate the presence of cancer and/or other diseases, proteomic analysis of the NP corona is essential for providing specificity regarding cancer type and progression.

Following protein digestion, we performed label-free mass spectrometry (LC-MS/MS) analysis of the in vivo recovered NP coronas. The total number of identified proteins at D7, D14 and D18 in control and GB-bearing mice are shown in Supplementary Fig. 1c and d, respectively. It should be emphasised that for the proteomic analysis of the mouse plasma samples, our primary goal was not to identify potential biomarker proteins but rather to correlate subtle changes in the plasma proteome with those occurring at the tumour tissue level. Consequently, in discussing the mouse data, we do not claim the discovery of potential biomarkers but instead focus on identifying potential molecular pathways that could serve as sources for peripheral blood biomarkers. To this end, we included in our analysis all differentially abundant proteins (DAPs) with a $p$-value $< 0.05$, without applying False Discovery Rate (FDR) correction.

Our analysis revealed a total of 115 DAPs between GB-bearing and control mice, as early as D7 post-tumour inoculation (Fig. 2c). While

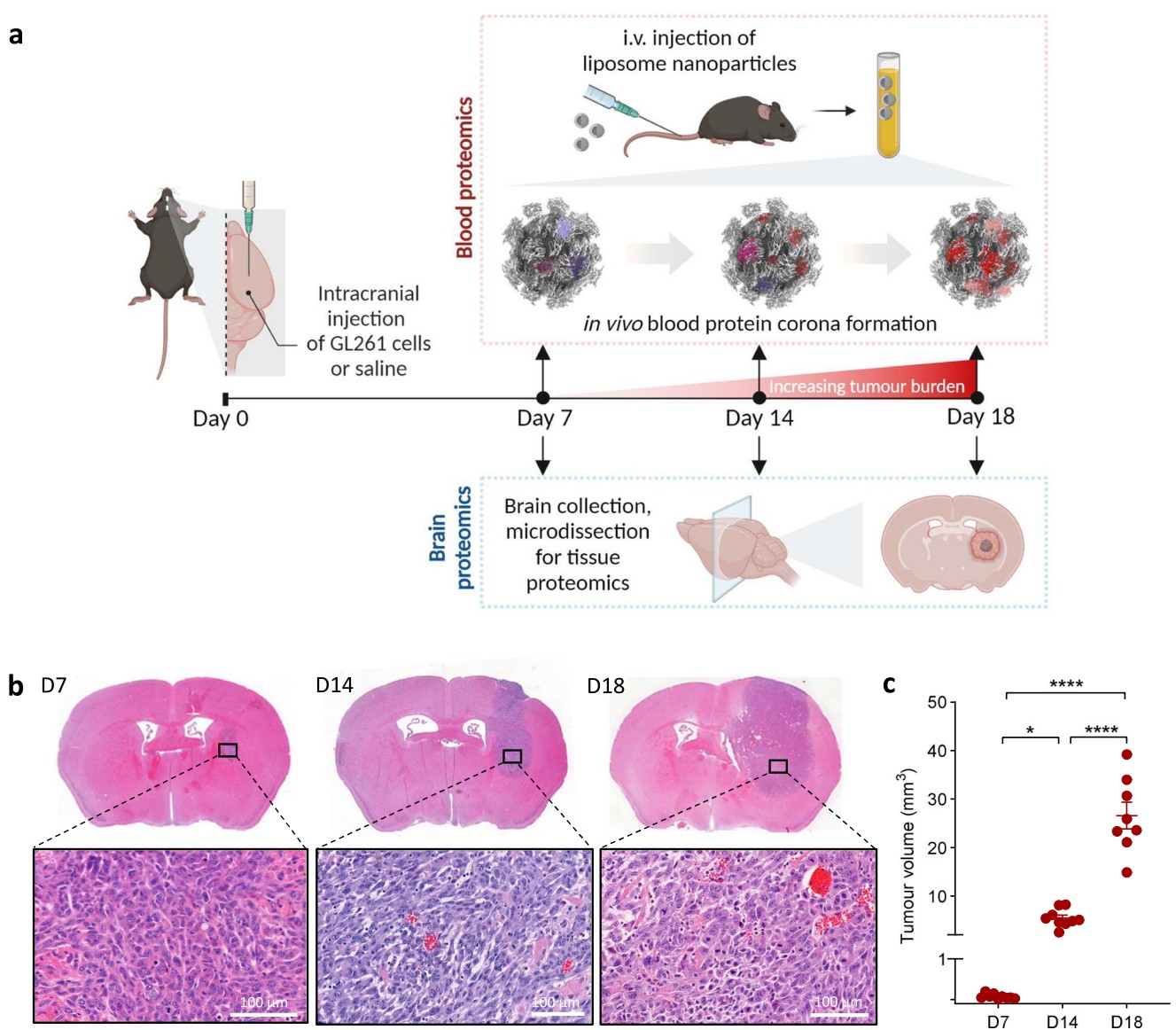

**Fig. 1 | Establishment of a longitudinal GL261 GB murine model. a** Overview of the in vivo study design enabling plasma and brain tissue proteomic profiling in GB tumour-bearing mice. A syngeneic GB murine model was established in C57BL/6 J female mice via intracranial injection of GL261 cells. Control mice underwent sham injection with saline. PEGylated liposome nanoparticles (NPs) were intravenously administered at days 7, 14, and 18 post-intracranial GL261 cell injection to allow protein corona formation. The corona-coated NPs were subsequently recovered from the blood circulation and purified to remove any unbound proteins. Brains were collected from control and tumour-bearing mice at all three-time points of

investigation. Created in BioRender. Hadjidemetriou, M. (2025) https://BioRender.com/z49a544. **b** Histological characterisation of GL261 tumours at day 7 (D7), day 14 (D14), and day 18 (D18) by haematoxylin and eosin (H&E) staining. **c** Quantification of the tumour volume in GB mice at days 7, 14, and 18 ($n = 9$ biological replicates for D7 and D14, $n = 8$ biological replicates for D18; error bars indicate mean $\pm$ SEM; *$p$-value = 0.0409 between D7 and D14, ****$p$-value < 0.0001 between D7 and D18, ****$p$-value < 0.0001 between D14 and D18 by One-way ANOVA with Tukey's multiple comparison test). Source data for Fig. 1c are provided as a Source Data file.

the total number of identified proteins remained consistent between control and tumour-bearing mice (Supplementary Fig. 1c, d), principal component analysis (PCA) demonstrated adequate separation of the proteomic data sets at all the three-time points (Supplementary Fig. 1e–g). Moreover, the number of DAPs increased from 206 at D14, to 417 at D18 (Fig. 2c). Analysis of the top 20 most abundant proteins identified at D7, D14, and D18 indicates similar profiles between control and GB-bearing mice, suggesting that the observed increased amount of corona proteins in GB-bearing mice was not due to a few proteins disproportionately affecting the overall protein composition (Supplementary Fig. 1h).

The majority of DAPs were upregulated in tumour-bearing mice (Fig. 2d, e), suggesting a progressive shedding of tumour-related

proteins into the bloodstream with increased tumour burden. The small overlap in DAPs at the three-time points (Fig. 2c) suggests distinct proteomic changes at different stages of tumour progression, highlighting the importance of longitudinal, rather than cross-sectional, biomarker discovery studies to fully capture the dynamic changes associated with GB progression.

**Longitudinal monitoring of the GB-specific plasma proteome**

Driven by the unmet clinical need to discover robust disease-monitoring biomarkers in GB, we investigated the temporal evolution of the in vivo-formed protein corona in GL261-bearing mice across the three different stages of tumour growth. As illustrated in Supplementary Fig. 2a, PCA of the proteomics data at the three time points,

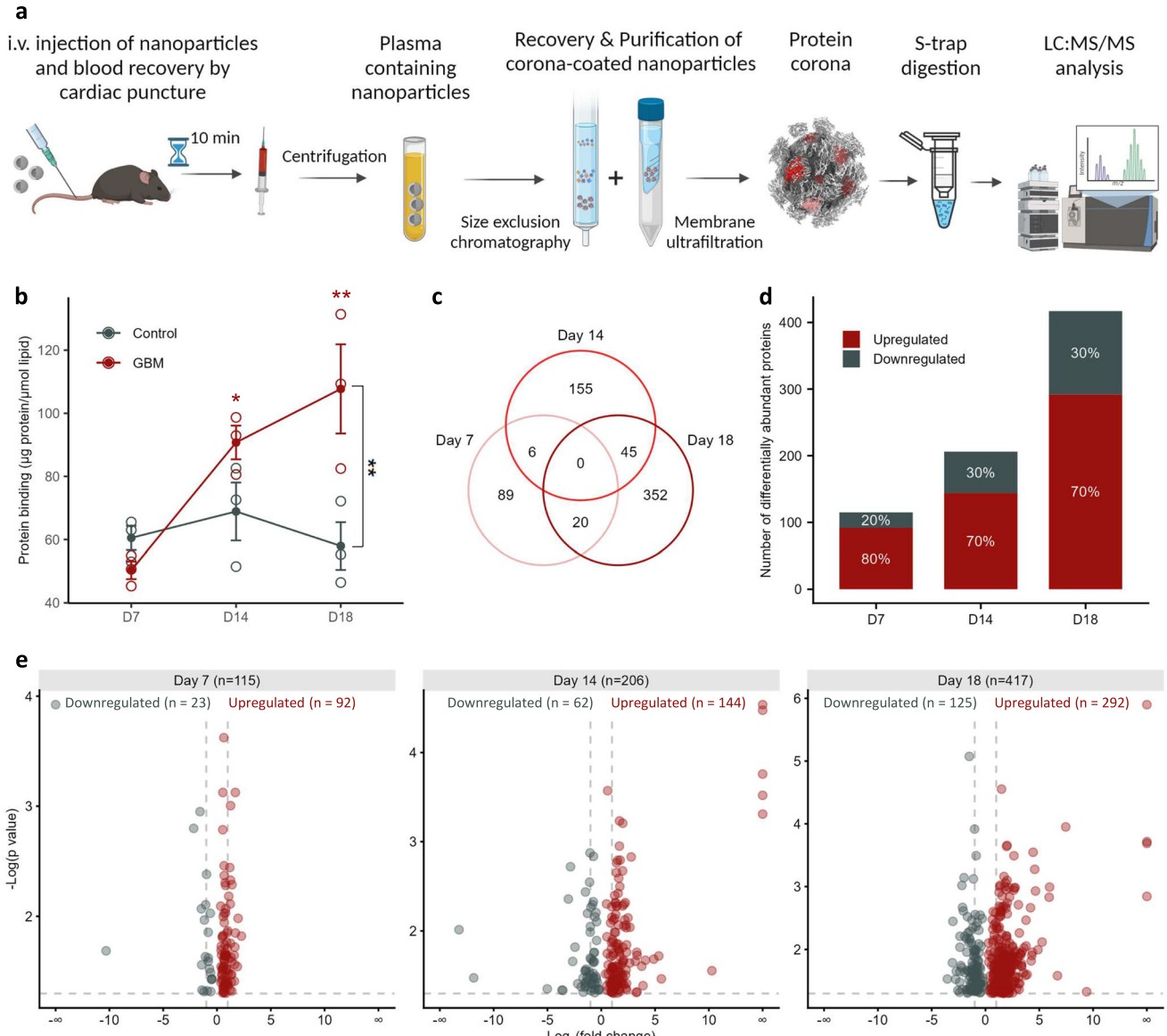

**Fig. 2 | Nanotechnology-enabled plasma proteomics in the GL261 mouse model of GB. a** Schematic overview of the in vivo plasma proteomics workflow. Following the intracranial injection of GL261 glioma cells (for GB-bearing mice) or saline (for control mice), C57BL/6 J female mice were intravenously administered liposome NPs at days 7, 14, and 18 post-tumour implantations. Blood-circulating NPs were subsequently recovered, and corona-coated NPs were purified prior to LC-MS/MS analysis. Created in BioRender. Hadjidemetriou, M. (2025) https://BioRender.com/g08d103. **b** The total amount of protein adsorbed onto the surface of liposome NPs was quantified and expressed as protein binding value (μg of protein/μmol lipid). Error bars indicate mean ± SEM of $n = 3$ biological replicates (plasma pooled from $n = 5$ mice for each biological replicate; * $p$-value = 0.0414 between D7 GBM and D14 GBM; ** $p$-value = 0.009 between D7 GBM and D18 GBM; One-way ANOVA with Tukey's multiple comparisons test between three-time points within the GB group; ** $p$-value = 0.0028 by Sidak's multiple comparisons between the control and GB groups at D18 time point. **c** The Venn diagram illustrates the number of common and unique differentially abundant proteins (DAPs) identified at D7, D14, and D18 time points (proteins with a $p$-value < 0.05). Statistical comparisons of relative protein expressions between corona proteomes from tumour-bearing mice and control mice were conducted using Progenesis QI for proteomics software (v. 3.0; Nonlinear Dynamics). **d** Bar graph reports the number and percentage of upregulated and downregulated DAPs identified through the analysis of the protein coronas formed in GB and healthy control mice ($n = 3$ biological replicates; pooled plasma from $n = 5$ mice per biological replicate). **e** Volcano plots display the relationship between fold change (shown in $x$-axis) and statistical significance (shown in $y$-axis) of the DAPs (with one-way ANOVA $p$-value < 0.05) at D7, D14 and D18 time points. The comprehensive lists of DAPs are provided in Supplementary Data 1–3. Source data for Fig. 2b are provided as a Source Data file.

revealed distinct plasma proteomic profiles corresponding to each stage of tumour growth. Moreover, our findings indicated a gradual decrease in the abundance of the 115 DAPs identified at D7 as the tumour burden increased (Fig. 3a). In contrast, the prevailing pattern among the 417 DAPs identified at D18 followed a gradually upregulation trend during the time course from low to high tumour burden (Fig. 3b). The opposite kinetics observed in these two protein groups suggested the presence of potentially distinct underlying mechanisms

during the establishment and growth of the tumour within the host brain.

Further comparative analysis of the NP corona proteomes in GB-bearing mice at different time points revealed 407 DAPs between D7 and D14, and 532 DAPs between D14 and D18 (Fig. 3c, d). In addition, we identified a group of 116 proteins with differential abundance across all three time points (filled dots in the volcano plots of Fig. 3c, d). These 116 DAPs were grouped into six clusters based on their longitudinal

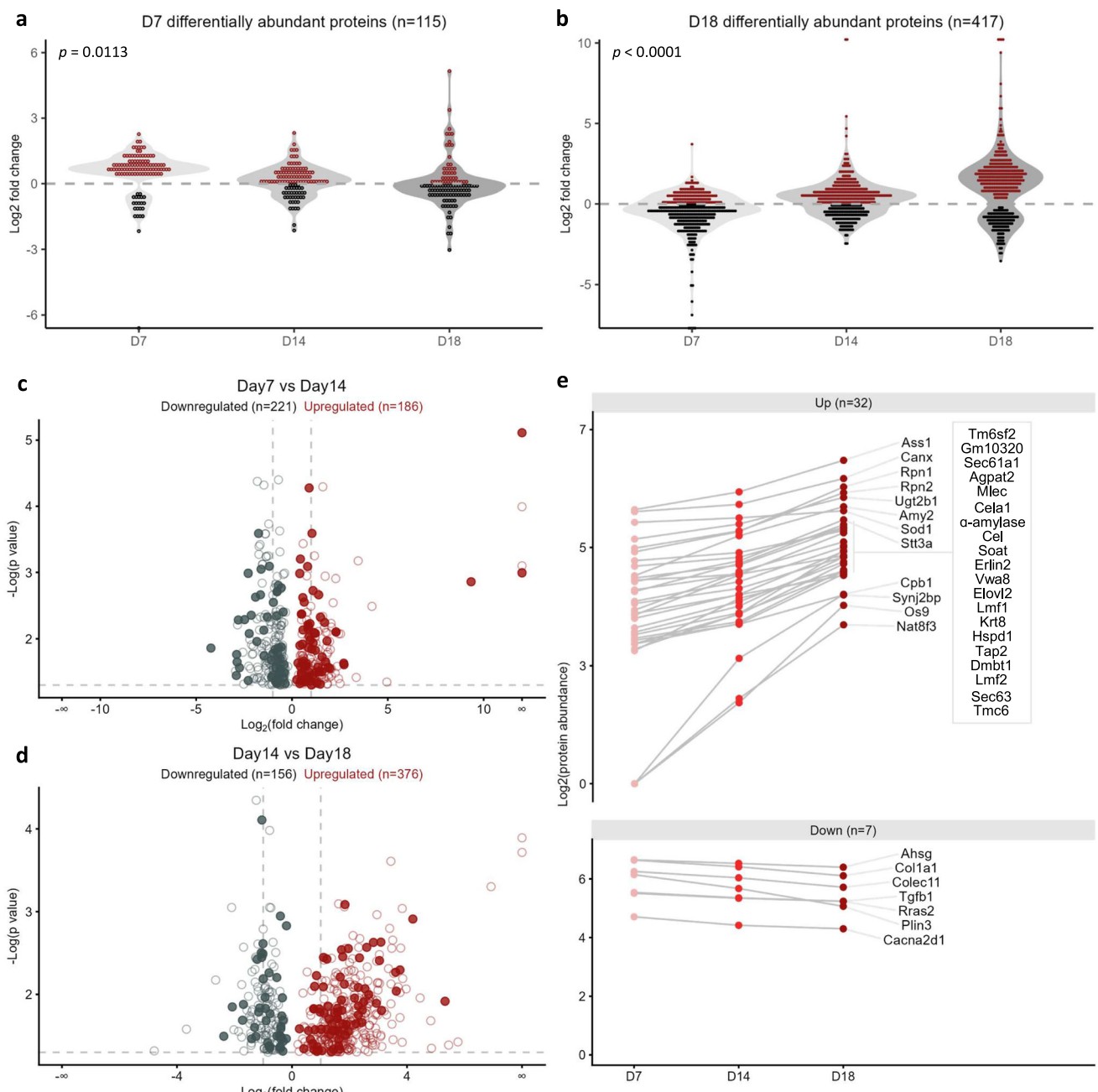

**Fig. 3 | Longitudinal monitoring of the GB-specific plasma proteome in GL261-bearing mice. a, b.** The longitudinal fluctuation in the fold change values of the $n = 115$ and $n = 417$ proteins identified as DAPs between GB and control mice at D7 and D18, respectively ($p$-value = 0.0113 and $p$-value < 0.0001 by one-way ANOVA with Geisser-Greenhouse correction, respectively). **c, d** Volcano plots display the relationship between fold change and significance for the DAPs (with one-way ANOVA $p$-value < 0.05) identified between D7 vs D14 and D14 vs D18 time points ($n = 3$ biological replicates; pooled plasma samples from $n = 5$ mice per biological replicate). DAPs identified as a result of mice aging between the three time points were excluded from the analysis by analysing the plasma proteome of sham-injected mice. The 116 common proteins between D7 vs D14 and D14 vs D18 GB mice highlighted in filled dots. The complete lists of DAPs identified between D7 vs D14 and D14 vs D18 time points are provided in Supplementary Data 4 and 5. **e** Long-itudinal kinetics in the abundance of the 32 upregulated ('Up') and $n = 7$ down-regulated ('Down') DAPs that displayed differential abundance between D7 vs D14 and D14 vs D18 time points (Supplementary Data 6).

abundance fluctuations (Supplementary Fig. 2b). Proteins exhibiting progressively increasing or decreasing plasma levels (Clusters 4 and 6) as tumour burden increased hold potential as biomarkers for GB monitoring and highlight the need for future longitudinal biomarker validation studies. Among these proteins (shown in Fig. 3e and Supplementary Fig. 2c), the serum biomarker alpha-2-HS-glycoprotein (AHSG) has been previously proposed to have high prognostic value in human GB[26,27]. Overall, our findings suggest that protein corona composition mirrors alterations in the plasma proteome in response to

brain tumour growth and underscore the need for longitudinal biomarker discovery studies over cross-sectional approaches[27–29].

**Integrated analysis of the plasma and tumour tissue proteomes**
Given the disruption of the blood-brain barrier (BBB) in GB and its consequent increased permeability, the detection of circulating tumour material in peripheral blood has been suggested[30,31]. However, the extent of overlap between GB-associated protein dysregulation at the tumour tissue and plasma proteome levels remains largely

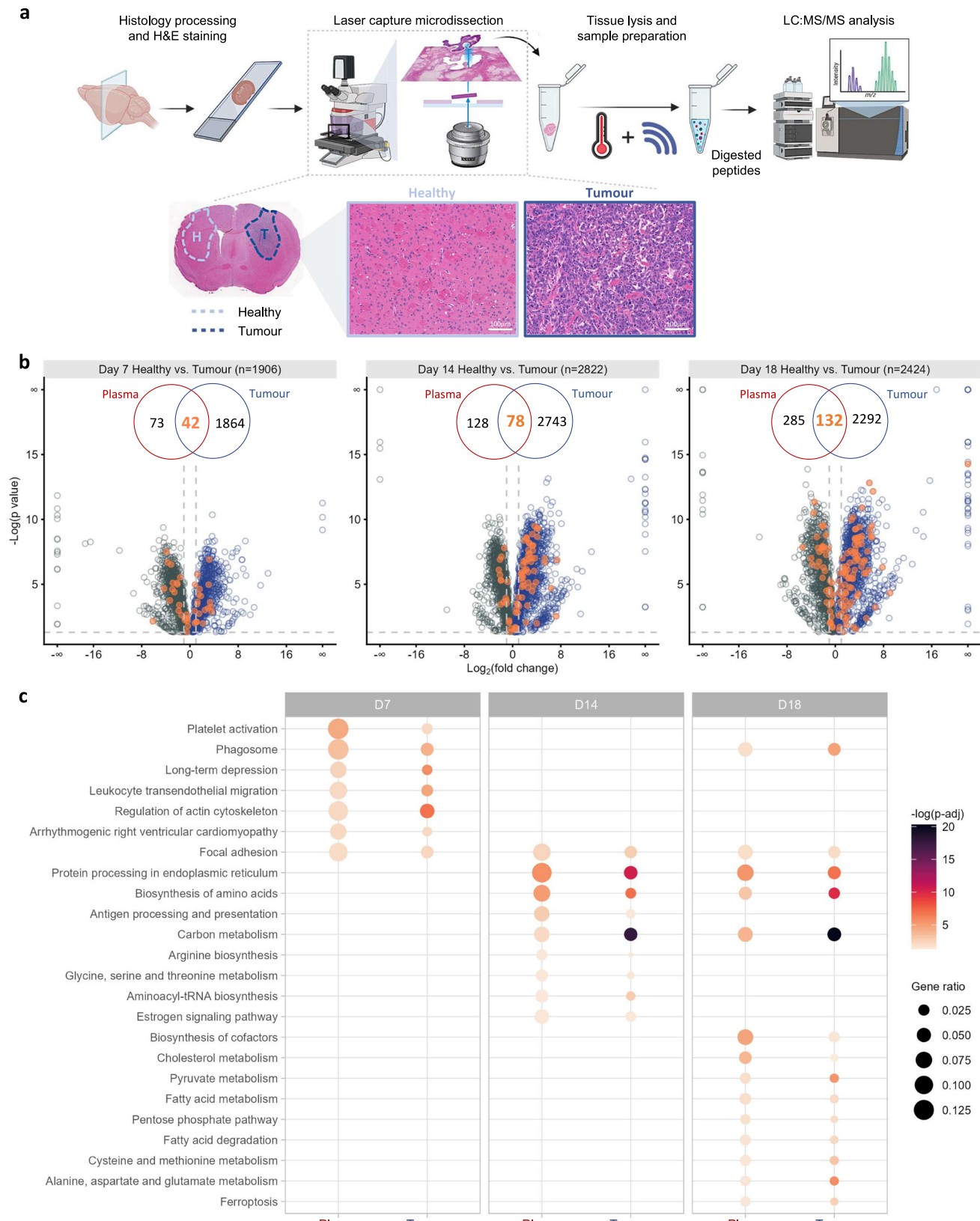

unexplored. To bridge this gap and establish a correlation between longitudinal changes in the plasma proteome with GB growth in the brain, we developed an integrative plasma-to-tumour tissue proteomics analysis workflow.

To allow spatially resolved and unbiased proteomics of formalin-fixed paraffin-embedded (FFPE) brain tissue samples, we implemented a laser-capture microdissection (LCM) coupled with LC-MS/MS protocol. This workflow enabled the proteomic comparison between the GL261 tumour tissue and 'healthy' brain tissue microdissected from the contralateral (non-injected) hemisphere (Fig. 4a). As shown in Fig. 4b and Supplementary Fig. 3a, differential protein abundance analysis revealed a notable number of DAPs between 'healthy' and 'tumour' tissues at all

**Fig. 4 | Integrative analysis of the plasma and tumour tissue proteomes in GL261-bearing mice. a** Schematic overview of the laser-capture microdissection (LCM)-coupled LC-MS/MS workflow employed for the proteomics analysis of the brain tissue samples. FFPE-preserved brains (*n* = 6 biological replicates/time point) were histologically sectioned and stained with haematoxylin and eosin (H&E) to reveal pathological characteristics of the tissue sections. Using a laser capture microscope, a total volume of 0.1 mm³ brain tissue was dissected from the tumour-bearing hemisphere and the 'non-injected' lateral hemisphere. Equivalent volumes of brain tissue from the sham injection site and the lateral hemisphere of the same brain were microdissected from sham-injected control mice. Microdissected brain tissue samples were lysed and subjected to S-trap protein digestion prior to their analysis by LC:MS\MS. Created in BioRender. Hadjidemetriou, M. (2025) https://BioRender.com/k25w778. **b** Volcano plots display the relationship between fold change and significance for the DAPs identified through comparison of healthy and tumour tissues. All proteins with a one-way ANOVA *p*-value < 0.05 are shown, coloured for upregulation shown in blue and downregulation in black. Any DAPs also identified in sham-injected mice (Supplementary Fig. 3c and Supplementary Data 7−9) were excluded from the lists. DAPs that were common between plasma and brain tissue are highlighted in orange. The full list of DAPs is shown in Supplementary Data 10−12. Venn diagrams illustrate the overlap between the plasma and tumour tissue identified DAPs at D7, D14 and D18 (Full list in Supplementary Data 13). **c** Dot plot presents the common KEGG pathways identified through independent enrichment analyses of plasma and tumour tissue DAPs at D7, D14 and D18, using a significance threshold of adjusted *p*-value < 0.05 by one-sided Fisher's exact test. Pathways are ranked according to the adjusted *p*-value within each time point. The colour of the dots represents the adjusted *p*-value, and the size of the dots indicates the gene ratio (genes involved/total number of genes). Source data for Fig. 4c are provided as a Source Data file.

three-time points of investigation (*n* = 1906 at D7, *n* = 2822 at D14, and *n* = 2424 at D18). A total of 1227 DAPs were found to be common across all different stages of tumour growth (Supplementary Fig. 3a).

We next investigated whether there were any commonly identified DAPs between the plasma and tumour tissue proteomics analyses. Remarkably, we found that over 30% of plasma DAPs identified at each of the three stages of tumour growth were also identified to be differentially abundant in the tumour tissue in comparison to control brain tissue (Venn diagrams of Fig. 4b). Ingenuity Pathway Analysis (IPA) of the plasma-tumour tissue overlapping DAPs (Supplementary Fig. 3b) revealed an increasing number of intracellular proteins as the tumour progresses, suggesting greater shedding of tumour-derived materials into the bloodstream. This finding aligns with elevated Pb values (Fig. 2b), and the upregulation kinetics trend (Fig. 3b, e) observed with increased tumour burden. It should be noted that equivalent volumes of brain tissue from sham-injected control mice were also subjected to LC:MS/MS analysis to account for any changes in the brain tissue proteome caused by the intracranial injection (Supplementary Fig. 3c).

As illustrated in Supplementary Fig. 3d, our data identify DAPs that exhibit both consistent and opposite dysregulation trends between plasma and tumour tissue. Among the overlapping DAPs, 45.2, 61.5, and 42.4% showed increased abundance in both blood and tumour tissues at D7, D14, and D18, respectively. Additionally, the percentage of overlapping DAPs consistently downregulated in both plasma and tumour tissues increased from 0% at D7 to 3.8% at D14 and 12.1% at D18. Notably, the percentage of proteins that were downregulated in plasma but upregulated in the tumour increased from 2.4% at D7 to 18% at D14 and 18.2% at D18, suggesting a possible movement of proteins from blood to tumour tissue to support the brain's high metabolic activity. Finally, 52.4%, 16.7%, and 27.3% of common DAPs were found to be upregulated in blood but downregulated in tumour tissue, indicating that multiple mechanisms involved beyond the commonly accepted release of materials from expanding tumours into the bloodstream. This data is in agreement with the two opposite kinetic trends observed during the establishment and progression of GB, suggesting a systemic response to the inoculation of the tumours at earlier stages (Fig. 3a, b).

Given the significant overlap between the plasma- and tumour-identified DAPs, we then examined whether the NP-enriched plasma proteome could reflect the tumour tissue proteome at a molecular pathway level. To gain a mechanistic understanding of the systemic response to tumour growth in the brain, we analysed the Kyoto Encyclopaedia of Genes and Genomes (KEGG) pathways enriched throughout the transition from low to high tumour burden. As depicted in Fig. 4c, the number of shared pathways over-represented by both plasma and tumour DAPs increased proportionally with tumour growth. The complex kinetics of the plasma protein signatures (observed in Fig. 3a−e), were also reflected at the molecular pathway analysis level, signifying a dynamic transition of underlying molecular

mechanisms at different stages of tumour growth (Fig. 4c). At the earliest time point (D7) seven pathways were commonly over-represented by plasma and tumour DAPs, including platelet activation, phagosome and regulation of actin cytoskeleton pathways. At the D14 time point, protein processing in the endoplasmic reticulum and biosynthesis of amino acids were among the most enriched shared pathways identified. Finally, the pathways commonly enriched in both plasma and tumour tissue at the D18 time point, underscored a highly metabolic activity induced by the rapid proliferation of tumour cells[32,33].

Together these findings suggest that the developed NP-enabled plasma-to-tumour tissue integrative proteomics pipeline could facilitate a molecular mechanism-guided biomarker discovery approach. In this context, we aimed to map six key KEGG molecular pathways with common plasma and tumour DAPs (Supplementary Fig. 3e). Specifically, the mapped pathways were the phagosome, protein processing in the endoplasmic reticulum, focal adhesion, regulation of actin cytoskeleton, carbon metabolism and biosynthesis of amino acids. A total of 46 DAPs were linked to the six pathways, the majority of which were found to be upregulated in the plasma of tumour-bearing mice (Supplementary Fig. 3f).

The protein processing in the endoplasmic reticulum was found to be the largest node connected to a total of 20 DAPs (Supplementary Fig. 3e). Notably, focal adhesion stood out as the only plasma-to-tumour common pathway identified at all three-time points (Fig. 4c and Supplementary Fig. 3e, f), emphasising its potential involvement in the growth and progression of GB. As illustrated in Supplementary Fig. 3f, among the proteins associated with the focal adhesion pathway, vinculin (Vcl), zyxin (Zyx), integrin subunit alpha 6 (Itga6), and Ras-related protein Rap-1b (Rap-1b) were all found to be upregulated in the plasma as early as D7 post-inoculation of GL261, suggesting their potential utility of proteins involved in the focal adhesion pathway for the early detection of low-burden GB. Consistent with our findings, the focal adhesion pathway was also previously identified as the most enriched pathway in proteomics analysis of longitudinal serum samples obtained from GL261-bearing mice[27].

While the phagosome, protein processing in the endoplasmic reticulum, biosynthesis of amino acids and carbon metabolism pathways could provide biomarkers of GB progression, the focal adhesion- and regulation of actin cytoskeleton-associated proteins could also potentially detect low-burden GB (Supplementary Fig. 3e, f). Among the 46 DAPs linked to the six investigated pathways, the following seven showed progressive up- or down-regulation with increasing disease burden (Fig. 3e and Supplementary Fig. 2c): argininosuccinate synthase (Ass1); calnexin (Canx); dolichyl-diphosphooligosaccharide protein glycosyltransferase subunits 1 and 2 (Rpn1 and Rpn2); Ras-related protein R-Ras2 (Rras2); STT3 oligosaccharyltransferase complex catalytic subunit A (STT3a); and SEC61 translocon subunit alpha 1 (Sec61a1). Canx, Rpn1, Rpn2, STT3a, and SEC61 are all associated with the protein processing in the endoplasmic reticulum pathway, while

Rras2 is linked to the regulation of actin cytoskeleton and Ass1 is part of the protein network related to the biosynthesis of amino acids pathway (Supplementary Fig. 3e, f). Moreover, Canx and Sec6a1 proteins are also linked to the phagosome pathway, which was found to be shared between the D7 and D18 time points, in both plasma and tumour tissue proteomes (Supplementary Fig. 3e, f). Notably, six out of the above-mentioned seven proteins were also found to be upregulated in the tumour tissue (Supplementary Fig. 3f), which may suggest their gradual increased secretion from the tumour mass and into the blood circulation.

Overall, our results underscore the capability of the Nano-omics integrative proteomics workflow to uncover GB-specific molecular pathways as potential sources for peripheral blood biomarkers that reflect brain tumour pathophysiology. Considering the limitations of syngeneic mouse models, further validation work using somatic models initiated by different sub-type relevant driver mutations would better represent the clinical heterogeneity and could enhance our mechanistic understanding of GB progression[34,35].

## Validation of the plasma-to-tumour tissue integrative workflow in human GB

While the molecularly richer in vivo NP corona has demonstrated its potential in facilitating the discovery of blood biomarkers in preclinical models[11,12], the ex vivo NP corona has been employed for the analysis of human plasma samples[13,14]. The disease specificity of the biomolecule corona and the concept of 'personalised corona', has been elucidated in previous studies through the analysis of human plasma samples collected from patients with various underlying conditions[15,16,36]. Although comparing protein coronas formed in 'healthy' and 'diseased' states can facilitate the discovery of multiple disease-associated proteins, it remains challenging to identify potential biomarker proteins that are solely linked to the primary disease without being influenced by individual factors such as sex, age and comorbidities. In the case of cancer biomarker discovery, when resection of the tumour is part of the clinical regimen, correlation analysis between the plasma and resected tumour tissue proteomes has the potential to address this challenge by selecting the most biologically relevant biomarkers and/or exploring deeper onto the underlying molecular mechanisms[17].

To provide 'proof-of-principle' validation of our plasma-to-tumour tissue integrative proteomics workflow in a clinical setting, we employed the ex vivo Nano-omics pipeline to analyse matched plasma and tumour tissue samples from n = 10 patients with high-grade gliomas (Fig. 5a and Supplementary Fig. 4a, b). It is important to note that the analysis of tumour-free control brain tissue was not feasible due to ethical and practical constraints. Therefore, the proteomics analysis of human tumour tissue was intended to refine the list of potential blood biomarkers, rather than serving as a standalone biomarker analysis. This limitation underscores the importance of preclinical biomarker research, where more controlled experimental conditions are possible.

To compare the pre-operative plasma proteome of GB patients with age- and sex-matched healthy controls (n = 10), we characterised the ex vivo formed protein corona around liposome NPs (same liposomes as those used for in vivo NP corona characterisation in GL261-bearing mice). In addition, we applied LCM-coupled LC:MS/MS to analyse matched snap-frozen tumour tissue samples obtained from the same GB patients (by surgical resection) (Fig. 5a and Supplementary Fig. 4b).

The comparative analysis of the NP protein corona samples unveiled a total of 321 plasma proteins exhibiting differential abundance between GB patients and healthy controls, (n = 216 upregulated; n = 105 downregulated), of which 272 exhibited a fold change > 2 (Fig. 5b and Supplementary Data 14). Only proteins with an FDR-adjusted p-value (q-value) < 0.05 were included in our human plasma proteomics analysis. Among the 321 DAPs, ~44% (n = 140 proteins

highlighted in Fig. 5b) were also identified by proteomics profiling of the resected tumour tissue (Supplementary Data 15). The significant overlap observed between the plasma and tumour tissue proteomes in both mice and humans (Figs. 4b and 5b) further signifies the potential of integrative plasma-to-tumour tissue proteomics approaches for GB-specific biomarker discovery.

We then conducted a cross-species comparison of the GB-specific plasma and tumour tissue proteomes, in order to provide a data analysis workflow (illustrated in Supplementary Fig. 5a) that facilitates the identification of the most biologically relevant potential biomarker proteins. Comparison of the plasma DAPs identified in humans and mice revealed a total of 61 commonly identified DAPs (Supplementary Fig. 5b). Considering the longitudinal fluctuation in the abundance of plasma proteins identified with GB progression in the GL261 model (Fig. 3), we chose to include all time points of investigation in our mouse-to-human correlation analysis. As shown in Supplementary Fig. 5b, the D18 time point exhibited a higher number of shared DAPs (n = 38) between mouse and human species compared to D7 (n = 19) and D14 (n = 23). This outcome was anticipated, given the higher number of mouse DAPs identified at D18 (Fig. 2e). Interestingly, 16 out of the 61 common DAPs in plasma were also identified through the analysis of the tumour tissue proteome in both mice and humans (Supplementary Fig. 5c). A closer examination of the fold-change trends for all 61 DAPs, revealed a 100% correlation in the dysregulation patterns between mice and humans at D7, which decreased to 65.2% at D14 and 52.6% at D18 (Supplementary Fig. 5d). These findings highlight the importance of considering longitudinal data when conducting cross-species comparisons.

To elucidate the molecular mechanisms underpinning human GB, a KEGG pathway analysis was conducted. The following four functional groups of KEGG pathways were found to be over-represented by both human plasma DAPs and tumour tissue-identified proteins: cytoskeleton and extracellular matrix, immune system, metabolism, and cellular and molecular signalling (Fig. 5c). Among all the pathways associated with each of the four functional groups, the regulation of actin cytoskeleton, focal adhesion, platelet activation, leukocyte transendothelial migration, biosynthesis of amino acids, carbon metabolism and phagosome were found to be shared between both human and mouse species (Fig. 4c and Fig. 5c). According to the longitudinal pathway analysis conducted in the GL261 mouse model (Supplementary Fig. 3e, f), while the cytoskeleton and extracellular matrix related pathways may serve as potential sources of biomarkers for detecting low tumour burden, the metabolic and cellular/molecular signalling pathways have the potential to yield biomarkers indicative of GB progression.

Subsequently, we linked the seven cross-species shared pathways with DAPs commonly identified in human plasma and tumour proteomes (Fig. 5d). As a result, a panel of 48 proteins was mapped according to their involvement in the respective pathways (Fig. 5d). The regulation of actin cytoskeleton was identified as the most enriched pathway linked to a total of 21 DAPs. Cross-species correlation analysis revealed that, out of the 48 human DAPs mapped to the seven pathways, 17 were also found to exhibit differential abundance in the plasma of GL261 tumour-bearing mice, with 10 of these also identified in the mouse tumour tissue (Fig. 5d and Supplementary Fig. 6).

As shown in Supplementary Fig. 6, 45 out of the aforementioned 48 DAPs were found to be upregulated in human plasma, indicating the activation of the seven-enriched pathways at the tumour tissue level and an increased shedding of GB-specific proteins into the blood circulation as a result. Within this group of proteins, Phosphofructokinase, Liver Type (PFKL), Von Willebrand Factor (VWF) and Tubulin beta-4B chain (TUBB4B) were ranked among top 5 most significant plasma DAPs between GB patients and healthy controls, exhibiting a log2 fold change > 10 or -log p-value > 6 (Fig. 5b). Notably,

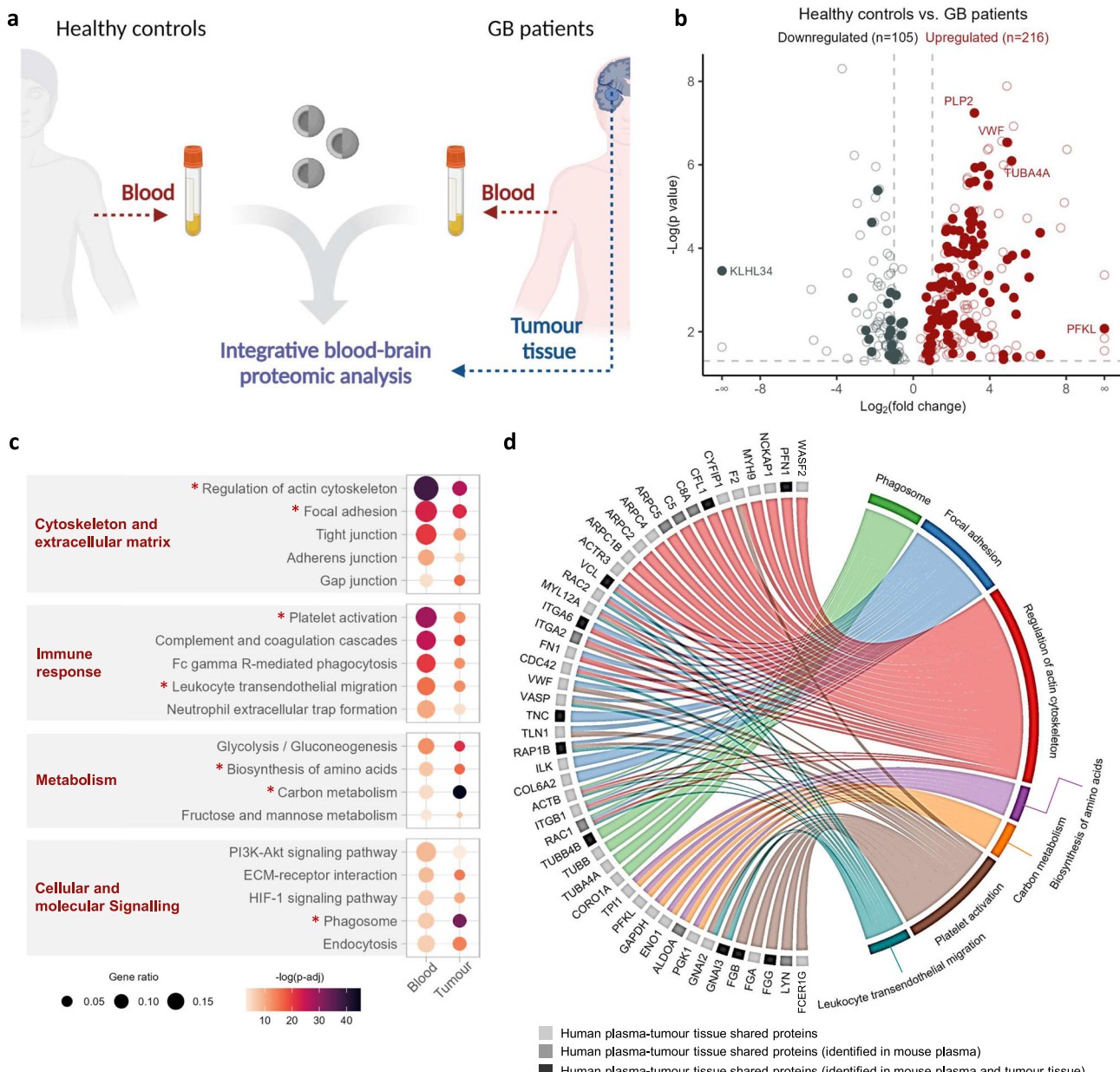

**Fig. 5 | Human clinical validation of the plasma-to-tumour tissue integrative proteomic analysis pipeline. a** Schematic representation outlining the human clinical cohort study design. Plasma samples obtained from pre-operative high-grade glioma patients were matched with healthy controls, both then formed ex vivo coronas with liposome NPs. Snap-frozen tumour tissue from the same GB patients during tumour surgical resection were cryo-sectioned and annotated by a clinical pathologist before laser capture microdissection (Supplementary Fig. S4b). Both corona and tumour tissue samples were subjected to LC-MS/MS analysis. Created in BioRender. Hadjidemetriou, M. (2025) https://BioRender.com/k40i192. **b** Volcano plot illustrating the relationship between fold change and significance for DAPs identified by comparing GB patient-corona with healthy control-corona ($n = 10$ biological replicates). A total of $n = 321$ proteins (with an FDR-adjusted one-way ANOVA $p$-value < 0.05) were found to be differentially abundant, of which $n = 272$ proteins exhibited a fold change > 2. Filled dots represent 140 proteins that were also identified by LC:MS/MS analysis in human tumour tissue. Protein names

with log2 fold change > 10 or -log($p$-value) 6 are displayed. Full lists of proteins are shown in Supplementary Data 14–16. **c** Dot plot presents common KEGG pathways identified through independent enrichment analyses of human plasma DAPs and tumour tissue proteins, using a significance threshold of adjusted $p$-value < 0.001 by one-sided Fisher's exact test. Pathways are classified into three groups according to the functional category in the KEGG pathway database. The colour of the dots represents the adjusted $p$-value, and the dot size indicates the gene ratio (genes involved/total number of genes). *Indicates the common enriched pathways between humans and mice. **d** Chord diagram connects the seven enriched pathways shared between humans and mice, with the $n = 48$ DAPs identified to be shared between the plasma and tumour tissue analyses. Human DAPs found to be shared with mouse plasma DAPs are represented in medium-grey colour. Human DAPs identified as common with mouse plasma and tumour tissue DAPs are depicted in dark-grey colour. The full list of the $n = 48$ DAPs is shown in Supplementary Fig. 6.

while several of the aforementioned 48 DAPs have been previously linked to GB progression, cell migration and invasion at a tumour tissue proteomic and/or transcriptomic level, only 11 proteins have previously been reported in the plasma of GB patients (Supplementary Fig. 6)[17,29,37–42]. Moreover, the Nano-omics integrative pipeline enabled

the identification of 16 newly discovered potential biomarker proteins for GB, which have not been previously reported at the plasma level in GB or any other cancer. Future work will investigate whether the combination of Nano-omics with a Data Independent Acquisition (DIA) bioinformatics pipeline,[43] further enhances proteome coverage in

comparison to the Data Dependent Acquisition (DDA) approach employed in this study.

It is widely accepted that a single biomarker is unlikely to provide the sensitivity and specificity required for early detection and monitoring, necessitating the need to develop a multi-protein biomarker panel. In the context of GB, patients often present with non-specific symptoms (e.g., headache), and a blood test would primarily serve to identify individuals suspected of having GB who should then undergo follow-up diagnostic Magnetic Resonance Imaging (MRI). The Nano-omics approach holds significant promise in this regard, enabling the discovery of multiple GB-associated proteins even at low tumour burden stages. While sex- and age-matched controls were used in our human cohort experiment, the small sample size ($n = 6$ males; and $n = 4$ females) limits our ability to draw conclusions regarding the sex-specificity of the candidate biomarkers identified. Therefore, the clinical utility of the above-mentioned DAPs as early detection and/or disease-monitoring biomarkers for GB necessitates future validation experiments in larger longitudinal patient cohorts, including a diverse group of patient control cohorts. This will require employing targeted immune approaches (e.g., Enzyme-Linked Immunosorbent Assay) to directly quantify the potential protein biomarkers in plasma (without the need for nanoparticle incubations), which is beyond the scope of this study.

Overall, our findings emphasise the importance of mechanism-centric approaches to biomarker discovery, facilitating the selection of highly specific blood biomarkers for subsequent clinical validation. Multi-dimensional proteomics combined with cross-species omics data correlation provides a valuable resource for discovering new biomarker candidates and revealing underlying molecular pathways[44,45]. The Nano-omics integrative platform developed in this study, provides an example of a technology springboard needed to yield comprehensive insight into early disease detection and disease monitoring and could potentially be deployed across a range of conditions affecting the central nervous system.

## Methods

### Animals
Female C57BL/6 J mice aged 12 weeks were purchased from Envigo and maintained under standard conditions at $22 \pm 2\,°C$, 40–60% humidity with standard 12 h light/dark cycles and free access to food and water. All procedures were performed in compliance with the United Kingdom Animals (Scientific Procedures) Act 1986 and local ethical approval under the Home Office project license P089E2E0A. Animals were coded to enable the blind evaluation of tumour volume.

### Human plasma and tumour tissue
Frozen plasma aliquots and tumour tissue from $n = 10$ GB patients ($n = 8$ primary and $n = 2$ recurrent) were provided by the Manchester Centre of Clinical Neuroscience following project review and ethical approval from at the Northern Care Alliance Research Collection (IRAS: 145069 – REC reference [19]NCABB001_001). Informed consent was obtained from all participants. Patient sample information is provided in Supplementary Fig. 4a. Sex- and age- (± 5 years old) matched human K2EDTA plasma control samples obtained from healthy volunteers were purchased from Cambridge Biosciences UK. All samples were frozen and stored at − 80 °C until use.

### GL261 tumour implantation
The murine GL261 cell line was obtained from the Leibniz Institute DSMZ, Germany. Cells were grown in RPMI-1640 media (R8758, Merck) supplemented with 100 UI/mL penicillin, 100 μg/mL streptomycin (P0781, Merck) and 10% foetal bovine serum (FBS, 10500064, Thermo Fisher Scientific) at 37 °C in 5% CO2. Intracranial implantation of tumour cells was performed as previously described[18]. Briefly, anaesthesia was induced with 3% isoflurane in 1.5 L/min O2 in a chamber, and

animals were then maintained under surgical depth anaesthesia. GB-designated mice received a stereotactically guided injection of 50,000 GL261 cells (50,000 cells/μL) into the striatum (− 2.3 mm lateral to bregma) using a 10 μl Hamilton syringe (1701RN, Hamilton) with a 26-gauge blunt needle (7768-02, Hamilton). Control mice received a saline (0.9% NaCl) injection of the same volume. The cell suspension or saline were slowly delivered (0.2 μL/min) at a depth of 2.4 mm from the pial surface. Analgesia (0.1 mg/kg buprenorphine) and fluid support (10 ml/kg saline) were administered, and animals were allowed to recover in a heated environment.

### Liposome preparation
HSPC: Chol: DSPE-PEG2000 (56.3:38.2:5.5) liposomes were prepared by thin lipid film hydration method followed by extrusion. To obtain liposomes with 25 mM final lipid concentration, lipids were firstly dissolved in chloroform: methanol mixture (4:1) in a total volume of 6 ml in a 100 ml round bottom flask. Organic solvents were evaporated using a rotary evaporator (Buchi, Switzerland) at 40 °C, at 150 rotations /min for 45 min under vacuum. The lipid film was hydrated with ammonium sulphate 250 mM (pH 8.5) at 60 °C to produce multi-laminar vesicles. Homogenous liposomes were produced by extrusion using LIPEX® 10 mL Thermobarrel Extruder (Evonik Canada Inc.). First, liposomes were passed five times through 800, 400 and 200 nm pore-size polycarbonate membranes (Whatman, VWR, UK). To achieve a smaller size, liposomes were passed through 400, 200 and 100 nm porous membranes for a further five times. Ammonium sulphate solution was exchanged using size exclusion chromatography (SEC) columns loaded with Sepharose CL-4B resin (Sigma-Aldrich) to obtain 25 mM liposome solution in HEPES (pH 7.4). The physicochemical properties of the liposome nanoparticles employed (Supplementary Fig. 1a) were characterised using the Zetasizer Pro instrument and automatically analysed by the ZS Xplorer software (Malvern Panalytical).

### In vivo administration of liposomes in GB tumour-bearing and healthy control mice
Liposomes were intravenously injected via the lateral tail vein (at a lipid dose of 0.125 mM/g body weight). This specific concentration was chosen because it is the equivalent lipid concentration to that of doxorubicin-loaded liposomes administered in preclinical studies to achieve a final doxorubicin dose of 5 mg/kg body weight. Corona-coated liposomes were subsequently recovered by cardiac puncture (10 min post-injection) from the blood circulation of GB tumour-bearing and sham control mice at 7 days, 14 days, and 18 days post-tumour implantations. For each time point, we conducted three biological replicates, with each replicate using pooled plasma from 5 mice (a total of $n = 15$ GB-bearing and $n = 15$ control mice were used for each time point of investigation). Approximately 0.2 - 0.3 ml of plasma was recovered from each mouse. This pooling was necessary to obtain a final plasma volume of 1 mL/biological replicate, which is required for the efficient purification and extraction of corona-coated liposomes and subsequent mass spectrometry analysis.

### Protein corona purification and quantification
An average of 0.5 mL blood containing corona-coated liposomes was collected in K2EDTA-coated vacutainer tubes (367839, BD) from each mouse. Plasma was then separated by centrifugation for 12 min at $1200 \times g$ at 4 °C after inverting the collection tubes to ensure the mixing of blood with EDTA. Pooled plasma from 5 mice was collected into Protein LoBind tubes (0030108116, Eppendorf) to reach a final volume of 1 mL for each biological replicate. Corona-coated liposomes were then separated from unbound plasma proteins by size exclusion chromatography followed by membrane ultrafiltration using Vivaspin 6 columns (10,000 MWCO, VS0602, Sartorious) and Vivaspin 500 columns (300,000 MWCO, VS0152, Sartorious)[8,22]. Proteins associated

with recovered liposomes were quantified using the Pierce BCA protein assay kit (Thermo Fisher Scientific). Lipid concentration was quantified by Stewart assay, and protein binding (Pb) values (µg of protein/µmol of lipid) were then calculated.

### Ex vivo liposome protein corona formation with human plasma

Liposome-plasma incubations and subsequent purifications were performed as reported previously[22]. In brief, 820 µL of human plasma and 180 µL of 12.5 mM PEGylated liposomes were incubated for 10 min at 37 °C, shaking at 250 rpm. The ex vivo corona was allowed to form using the same liposome concentration (2.25 mM) as that extracted in 1 mL of plasma from intravenously injected animals. Ex vivo protein corona was purified and characterised the same way as in vivo corona.

### Histology

**Mouse brain tissue.** Animals were perfused with PBS + 2% EDTA under terminal anaesthesia following a cardiac puncture. Mouse brains were collected and immerse-fixed in 4% paraformaldehyde for 24 h before being dehydrated and paraffin-embedded. Using a microtome, 5 µm coronal sections were taken within ± 2 mm from the injection site and mounted onto Superfrost Plus adhesion slides (J1800AMNZ, Epredia) and MMI membrane slides (50102, Molecular Machines & Industries). Both types of slides were then dewaxed with xylene and rehydrated with a series concentrations of ethanol solution before being stained with haematoxylin and eosin (H&E) using a Leica XL automated tissue stainer. Stained slides were sealed with coverslips and scanned as references for microdissection. Scanned images from $n = 8$–9 brains were used for tumour volume quantification. A total of $n = 6$ brains were used for tissue proteomic analysis at each time point. From sections mounted on MMI slides, equal volumes (0.1 mm³) of tumour tissue, and brain tissue from the same anatomical location in the contralateral hemisphere were microdissected. Equivalent volumes of brain tissue from the sham injection site and 'non injected' contralateral hemisphere were also microdissected from brains sections obtained from sham-injected control mice. To reach the required volumes, the above areas of interest from multiple continuous sections were dissected

**Human tumour tissue.** Snap-frozen human GB tissue samples ($n = 10$) were sectioned using a cryostat. 10 µm continuous sections were taken and thawed on both glass slides and MMI slides. All slides were then stained with haematoxylin and eosin (H&E) using a Leica XL automated tissue stainer (Leica). Stained slides were sealed with coverslips and scanned as references for microdissection. The tumour region within the human GB sections was annotated by a clinical neuropathologist to guide laser capture microdissection.

**Laser capture microdissection.** A MMI CellCut laser microdissection system (Molecular Machines & Industries) fitted on an Olympus IX83 inverted microscope (Olympus) and pooled into one isolation cap (50204, Molecular Machines & Industries). Tissue sections were dissected using the following parameters: 70 µm/sec laser velocity, 2500 µm laser focus, 2.5 mW laser power.

### Sample preparation for mass spectrometry

For protein corona samples, 10 µg of total protein was mixed with 10 µl of lysis buffer (50 mM TEAB with 5% SDS) and incubated at 4 °C for 1 h. Tissue samples were lysed as previously described with adjustments[46,47]. To reverse formalin-mediated protein cross-linking of tissue, a total volume of 0.1 mm³ dissected tissue was resuspended in 50 mM triethyl ammonium bicarbonate (TEAB) containing 5% (w/v) sodium dodecyl sulphate (SDS) with heating at 95 °C and shaking at 1400 rpm for 20 min, followed by heating at 60 °C and shaking at 1400 rpm for 2 h. Then, urea and dithiothreitol (DTT) were added to the samples to reach a final volume of 8 M and 5 mM, respectively and

incubated at 60 °C for 10 min. Samples were then transferred into a 130 µl AFA Fibre microTUBE (520077, Covaris) and sonicated using a LE220-Plus focused ultrasonicator (Covaris, UK) with the following settings: duration of 60 s, 500 W peak power, 20% duty factor, 200 cycle/burst, 100 W average power, repeated for 10 cycles (10 min total run time).

Lysed tissue and corona samples were then reduced by increasing the concentration of DTT by 5 mM and incubating at 60 °C for 10 min before further alkylation by 30 mM iodoacetamide (IAA) and incubating in the dark for 30 min. IAA was then quenched by the addition of DTT, and the solution was cleared by centrifugation at $14,000 \times g$ for 10 min. The cleared supernatant was transferred to a fresh tube and acidified by the addition of phosphoric acid to 1.2% (w/v). S-trap binding buffer (90% methanol in 100 mM TEAB, pH 7.1) was added to the samples to increase sample volume by 6-fold. Samples were then transferred to the S-Trap™ micro spin columns (ProtiFi) and washed once with methyl tert-butyl ether (MTBE)/methanol (10:3 in volume) solution followed by washes with S-trap binding buffer (2 additional washes for corona samples and 9 washes for tissue samples). In-column digestion by adding trypsin (1 µg trypsin in 20 µl for corona samples and 2 µg trypsin in 25 µl for tissue samples) and incubate at 47 °C for 1 h. The resulting peptides were eluted and subsequentially desalted in 96-well filter plate with 0.2 µm PVDF membrane (3504, Corning) using Oligo R3 resin beads (1-1339-03, Thermo Fisher Scientific) followed by 2 washes with 0.1% formic acid. Peptides were finally eluted with 0.1% formic acid in 30% acetonitrile and lyophilised using a SpeedVac vacuum concentrator (Thermo Fisher Scientific).

### Mass spectrometry and data analysis

Dried peptides were resuspended in 10 µl 0.1% formic acid in 5% acetonitrile and analysed by LC-MS/MS using an UltiMate 3000 Rapid Separation LC (RSLC, Dionex Corporation) coupled to a Q Exactive Hybrid Quadrupole-Orbitrap (Thermo Fisher Scientific) mass spectrometer. Peptide mixtures were separated using a gradient from 95% A (0.1% FA in water) and 5% B (0.1% FA in acetonitrile) to 18% B, in 34.5 min, 27% at 42.5 min, and 60% at 43.5 min, at a flow rate of 300 nL/min, using a 75 mm × 250 µm inner diameter 1.7 µM CSH C18 analytical column (Waters). Peptides were selected for fragmentation automatically by data dependant analysis (DDA). Data was required for 60 min in positive mode. All samples obtained from the mouse and human experiments were processed and run sequentially. Raw mass data were processed using Progenesis QI for proteomics software (v. 3.0; Nonlinear Dynamics). RAW files were imported into the software where automatic feature detection was performed. Following the automatic selection of the reference run, all other runs were aligned to the reference. Automatic processing with filters (maximum charge 5 of peak charge, relative quantitation using Hi-N method, $N = 3$ peptides to measure per protein) was specified. The resulting MS/MS peaks were exported and searched against the house mouse (*Mus Musculus*) proteome or the human (*Homo sapiens*) proteome (UniProt database) on a local Mascot server (v. 2.3.0; Matrix Science). The spectra were searched using the following parameters: Oxidation of methionine (M) was set as variable modifications and carbamidomethyl (C) was set as a fixed modification, trypsin digestion with one missed cleavage allowed peptide charges + 2 and + 3, precursor mass tolerance of 15 mmu and fragment mass tolerance of 8 ppm, ESI-QUAD-TOF for the instrument.

All raw protein corona data were compiled and subjected to a harmonised analysis, aligning with recent recommendations in nanoparticle corona analysis.[48] The produced XML file was imported into Progenesis QI v3.0 to map peptides to features with a peptide-spectrum match (PSM) score > 20, applying a 1% false discovery rate (FDR) filter to all significant ($p$-value < 0.05) PSM matches. The "between subject" experimental designs were established for pairwise comparisons between two or more groups by one-way ANOVA analysis. For biomarker candidate selection from the human clinical dataset,

type I errors were controlled by FDR-adjusted $p$-value ($q$-value) with a significance set at 0.05[49,50]. Results are reported as mean ± SD (normalised ion intensity score). The resulting protein lists with one-way ANOVA $p$-value, $q$-value, maximum fold change, and normalised protein abundance were exported from Progenesis and further analysed using Excel (Microsoft), R (v. 4.2.2), and RStudio (v. 2023.06.0, Posit Software)[51]. Plasma DAPs identified between the three-time points in sham-injected mice, as a result of mice aging, were excluded from the analysis (Fig. 3c, d). Similarly, the effect of intracranial injection on the brain tissue proteome was accounted for by excluding any DAPs identified between the injected and non-injected hemispheres of sham-injected mice (Fig. 4b).

### Integrative analysis of plasma and brain proteomic datasets

The 3-step integrative analysis workflow was developed. Firstly, the list of DAPs (proteins displayed an ANOVA $p$-value < 0.05 by comparing their abundance in diseased to control group) in the NP-enriched corona proteome was cross-examined with the list of DAPs identified in the tumour compared to healthy control, resulting in a list of plasma proteins with significantly altered expression in tumour tissue (common plasma-to-tumour proteins). In the case of the human clinical validation cohort, the tumour proteome was used for cross-examination due to the unavailability of healthy brain tissue. Human plasma and tumour DAPs were subjected to pathway overrepresentation enrichment analysis independently before overlapping pathways were identified. The "clusterProfiler" package from Bioconductor was used to perform overrepresentation analysis[52], where protein gene names were converted into entrezID using "org.Mm.eg.db" for mouse and "org.Hs.eg.db" for human to enable search. Gene ontology analysis of biological processes was performed using the function "enrichGO", and KEGG analysis was performed by the "enrichKEGG" function. For analyses with more than one lists the function "compareCluster" was used in conjunction with "enrichGO" or "enrichKEGG". A $p$-value cut-off = 0.05 was specified to output only statistically significant pathways. Common pathways identified in both data sets were visualised using the "ggplot2"[53], "enrichplot"[54], and "circlize" packages[55]. Finally, core molecular pathway networks were selected based on their longitudinal continuity in the mouse experimental setup or high statistical significance in the human clinical cohort. Common plasma-to-tumour proteins identified during the first step were then mapped to the selected core pathways.

### Statistical analysis and data visualisation

All statistical analyses and data plotting were performed using R (v. 4.2.2), RStudio (v. 2023.06.0, Posit Software)[51] and GraphPad Prism 9 software package (GraphPad Software). Illustrations and flowcharts were created with BioRender.com.

### Reporting summary

Further information on research design is available in the Nature Portfolio Reporting Summary linked to this article.

## Data availability

The data that support the findings of this study are available in the article and supplementary information files. The mass spectrometry proteomics data generated in this study have been deposited to the ProteomeXchange Consortium via the PRIDE[56] partner repository with the Accession code PXD060393. Source data is available for Figs. 1c, 2b, and 4c, and Supplementary Figs. 1b, h, and 3b in the associated source data file. Source data are provided in this paper.

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

## Acknowledgements

The authors acknowledge funding support from the United Kingdom Research and Innovation Engineering and Physical Sciences Research Council (UKRI EPSRC) 2D-Health Programme Grant EP/P00119X/1 (K. Kostarelos), the UKRI EPSRC International Centre-to-Centre grant EP/S030719/1 (K. Kostarelos), the Cancer Research UK (CRUK) International Alliance for Cancer Early Detection (ACED) EICEDAAP\100013 (M. Hadjidemetriou) and the CRUK Early Detection and Diagnosis Primer Award EDDPMA-Nov21\100027 (M. Hadjidemetriou). We would like to thank the Biological Mass Spectrometry (BioMS), Biological Services Facility, Histology Facility and Bioimaging Facility staff at the University of Manchester for their assistance and Ms. Irene Rebollido for her support in the preparation of liposomes. We would also like to thank the patients for their kind donation of blood and tumour tissue samples.

## Author contributions

X. Liu, performed the experiments, contributed to the methodology development, analysed the data, prepared the figures, and wrote the initial draft of the manuscript. H. Abmanhal-Masarweh, contributed to the preparation and characterisation of the liposome nanoparticles, conducted plasma control analysis and data analysis, and provided assistance with the in vivo experiments. O. Iwanowytsch, performed ingenuity pathway analysis of the proteomics data. E. Okwelogu, expanded and prepared the GL261 cells for the in vivo experiments and provided feedback on data analysis. K. Arashvand, provided assistance with the in vivo experiments and tissue collection. P. I. D'Urso and K. Karabatsou consented to the patients, collected pre-operative blood samples and collected the brain tumour tissue samples that have been investigated in this project. F. Roncaroli. contributed to the annotation of human tumour tissue in brain sections to guide microdissection and reviewed the manuscript. K. Kostarelos contributed to the conceptualisation of the study, ensured ethical compliance and overall responsibility for authorised in vivo work, provided intellectual input and feedback throughout the study and reviewed the manuscript. T. Kisby, contributed to the conceptualisation of the study, designed the

experiments, supervised the execution of the in vivo experimental work, contributed to the interpretation of the data, and reviewed the manuscript. M. Hadjidemetriou, contributed to the conceptualisation of the study, designed the experiments, supervised the execution of the proteomics experimental work and data analysis, contributed to the interpretation of the data, and prepared, reviewed and revised the manuscript.

## Competing interests

The authors declare no competing interests.
