## [Transparent Peer Review file · Nature Communications]

Plasma-to-Tumour Tissue Integrated Proteomics Using Nano-omics for Biomarker Discovery in Glioblastoma

Corresponding Author: Dr Marilena Hadjidemetriou

Version 0:

Reviewer comments:

Reviewer #2

(Remarks to the Author)

In this study, blood and tissue proteomics of mouse glioma model were investigated and integrated and analyzed using NP-enabled plasma proteomics method, and relatively informative experimental results were obtained, which are valuable for the understanding of glioma mechanism. However, there are still some concerns need to be addressed:

1. About the proteomics analysis method used in this study: NP-enabled plasma proteomics provides an alternative way to improve the coverage of blood proteomics, but its inherent limitations compromise its wide application, especially for real clinical studies. Injecting liposomes into the body and collecting enriched protein crowns for proteomic analysis after circulation is less suitable for human blood sample collection. The first issue is the safety of the material, and even though there are no significant health effects, injecting large amounts of liposomes into the bloodstream may cause problems related to elevated blood lipids. In addition, there may not be sufficient conditions to perform this process when collecting a large number of patient samples in a clinical setting. Therefore, the method can be used as a supplement to existing methods and is more suitable for animal level studies. However, it is also important to consider that the process of the method is cumbersome. The adsorption of proteins based on liposomes may have a certain preference, which can also lead to the loss of information about some proteins. In addition, individual differences in the animal body can lead to a significant difference in liposome uptake and liposome protein adsorption as well, which will make the impact of individual differences in cohort studies even more severe. In fact, in contrast, it may be more feasible to use methods such as DIA proteomics. It can both increase the coverage and quantitative accuracy of protein testing and is a non-invasive method. In a recent study, DIA proteomics has been able to detect on average more than 1300 proteins in plasma (J. Proteome Res. 2024, 23, 9, 3806-3822).

2. Regarding the innovative nature of this study: The authors claim that there are no reports of nanoparticle-enabled integrative blood-tissue proteomics, which underscores the innovative nature of this approach. There is nothing wrong with this statement. From this perspective, this study is somewhat innovative. However, the authors claim that "To our knowledge, this is the first study in the context of biomarker discovery for GB that attempts to integrate discovery proteomics at the plasma and brain tumour tissue levels" at the end of the manuscript. This statement is not correct. The reviewer note that studies have recently reported molecular mechanism studies and biomarker discovery for gliomas using the integration of blood proteomics and tissue proteomics, and that the entire study was conducted using human samples, which may have more clinical applications than the results based on animal experiments in this study (e.g., Sci. Adv. 2024, 10, eadk1721). In addition, this study used laser cutting to obtain both healthy and tumor tissue. While this allows for more precise sampling, it is not innovative, nor is it entirely necessary, except for proteomic analyses at the single-cell level.

3. The authors observed that the amount of bound protein enriched from the blood of mice became progressively greater as the tumor grew and was attributed to an increase in the material released into the blood stream as the tumor grew. A direct experiment should be added here to verify that the total protein concentration in the blood increases with tumor growth, rather than just speculation. If the total protein concentration does not change significantly, is it possible that the amount of certain proteins is increasing and the liposomes happen to adsorb those proteins preferentially. This is a question worth investigating, since the amount of protein binding is not just slightly increased, but almost doubled.

4. It is unreasonable that the authors did not use P-value correction for differential protein analysis in the mouse experiment. Although this could have shown more differential proteins, there could have been a large number of false positive results mixed in, which would have affected the reader's judgment of the experimental results. Sometimes more information is not better, because a large amount of mixed information will instead affect our understanding of the truth. Although t-test is a commonly used statistical method in differential protein expression detection to evaluate whether a protein is differentially expressed in two samples by combining variable data between samples. However, the test efficacy of the t-test is reduced due to the small sample size in this study and thus the less accurate estimation of the overall variance, and the number of false positives is significantly increased if the t-test is used multiple times. For example, when the p-value for a protein is less than 0.05 (5%), we usually assume that the expression of this protein is different in the two samples. But there is still a 5% probability that this protein is not a differential protein. Then we incorrectly deny the original hypothesis (no differential expression in the two samples), resulting in a false positive (5% probability of making a mistake). If the test is performed once, the probability of making a mistake is 5%; if it is performed 10,000 times, the number of mistakes made is 500, i.e., an additional 500 additional discrepancies are concluded (even if there are actually no discrepancies). To control for the number of false positives, we need to correct the p-value for multiple comparisons and raise the threshold. Moreover, for label-free methods, proteins with an absolute value of the multiplicity of differences >2 are generally selected for subsequent analysis.

5. In the analysis of tissue samples, equivalent volumes of brain tissue from sham-injected control mice were also subjected to LC-MS/MS analysis to account for any changes in the brain tissue proteome caused by the intracranial injection. But the results of the analysis and discussion of this data are not seen in the manuscript.

Reviewer #3

(Remarks to the Author)

Xinming Liu and colleagues, have revised the manuscript and accurately answered point by point to all the comments. Despite I acknowledge the effort in discussing some of the raised criticisms, the most important aspect about the specificity of the biomarkers composing the protein corona for GB patients has not been clarified. Although we all agree that a panel of biomarkers is better than a single marker and that we may witness the formation of "personalized corona", the protein composition of the corona should have a clear biological significance and specificity with respect to the disease studied to be useful in clinic.

The proteins mentioned remain clearly non-specific for glioblastoma. They are proteins already discovered in the past, associated to other cancers and disease.

The study remains interesting from a technical point of view, because it allows studying the formation and composition of the crown in relation to the size of the tumor (even if tumor size at the different times is not specified).

But it cannot be published with the title "Nano-omics Enables Plasma-to-Tumor Tissue Integrated Proteomics for Biomarker Discovery in Glioblastoma", because it does not in fact demonstrate the discovery of any new biomarker or group of markers contained in the corona specific for the GB.

Interesting would be at least to see a comparison between the candidate protein biomarkers identified in this study and those identified with a similar methodology in other cancer types (even if just at a literature level), proving some potential specificity.

Reviewer #4

(Remarks to the Author)

The manuscript presents an innovative approach utilizing in vivo protein corona analysis to monitor protein variations during the progression of glioblastoma in a mouse model. By comparing these findings with the protein corona composition of human plasma samples, the study identifies overlapping proteins that hold potential clinical value for the early detection of this aggressive disease. The integration of animal model data with human samples offers a promising pathway toward translational applications in glioblastoma diagnostics.

I have thoroughly reviewed the manuscript, including the reviewer comments and the authors' responses to the raised concerns. The additional experiments and explanations provided by the authors effectively address the initial critiques, enhancing the study's robustness and credibility. I am inclined to recommend publication of this insightful paper, pending the incorporation of the following revisions:

1- The manuscript should provide a more detailed description of the mass spectrometry techniques used. Specifically, it is important to clarify whether all raw protein corona data were compiled and subjected to a harmonized analysis. This approach has recently been shown to significantly improve the robustness and accuracy of the protein corona outcomes.

2- The authors should expand their discussion to consider the implications of personalized protein corona, taking into account factors such as sex, age, and other individual-specific variables. Understanding how these factors influence protein corona composition is essential for assessing the generalizability and applicability of the findings.

3- It is important to delineate whether the identified protein patterns are specifically associated with glioblastoma or if they are influenced by individual biological differences. This distinction is critical for validating the identified proteins as reliable biomarkers for glioblastoma. A practical approach would be to conduct protein corona analyses on plasma samples from both male and female subjects and examine the patterns of key proteins. Additionally, stratifying samples based on age groups or other relevant demographics could help isolate disease-specific protein signatures from those influenced by individual variability. Incorporating these analyses would not only enhance the study's depth but also provide a more comprehensive understanding of how personalized factors may affect protein corona composition. This, in turn, would refine

the proposed proteins as potential biomarkers, thereby increasing their clinical relevance and utility in glioblastoma detection.

Version 1:

Reviewer comments:

Reviewer #2

(Remarks to the Author)

The authors addressed some of the concerns through revisions, and the comments are:

1. The fact that using nanoparticles to enrich proteins in plasma or other body fluids, followed by proteomics analysis to improve the coverage of protein identification, is not an innovative research strategy and has been reported in many previous studies. The present study adopts the method previously used by the authors, in which liposomes are injected into mice and the enriched protein crowns are collected after circulation and then analyzed by proteomics, which might have some value in animal-level studies. However, such an approach is really not suitable for clinical applications. The authors also point out that for human plasma samples, a similar approach to that of other researchers, i.e., adding nanoparticles to centrifuge tubes for in vitro enrichment analysis, should be used. In summary, the novelty of this study is not in the methodology, but probably mainly in the biological application.

2. the authors observed that the amount of binding proteins enriched from the blood of mice became progressively greater as the tumor grew and was attributed to an increase in the amount of material released into the blood stream as the tumor grew. They also added an experiment confirming that the total protein concentration in the blood did not increase with tumor growth and concluded that the nanoparticle protein crowns were enriched for GB-specific proteins and/or extracellular vesicles. This conclusion is not very convincing. Why would nanoparticles specifically adsorb GB-associated increased proteins? Since there was a significant increase in protein, there must have been a significant decrease in protein because the total amount of protein did not change. Why did the GB-related decrease in protein not affect the total amount of protein adsorbed by the nanoparticles? It stands to reason that liposomes are not too specific for protein adsorption. It would be very strange if liposomes preferentially adsorbed the increased protein associated with GB. This experimental conclusion needs to be rigorously demonstrated.

Reviewer #4

(Remarks to the Author)

The authors have adequately addressed this reviewer's concerns.

REVIEWER COMMENTS

Reviewer #2 (Remarks to the Author):

In this study, blood and tissue proteomics of mouse glioma model were investigated and integrated and analysed using NP-enabled plasma proteomics method, and relatively informative experimental results were obtained, which are valuable for the understanding of glioma mechanism. However, there are still some concerns need to be addressed:

We would like to thank the reviewer for their thoughtful feedback. *Below, we provide a point-by-point response to address the reviewer's concerns:*

1. About the proteomics analysis method used in this study: NP-enabled plasma proteomics provides an alternative way to improve the coverage of blood proteomics, but its inherent limitations compromise its wide application, especially for real clinical studies. Injecting liposomes into the body and collecting enriched protein crowns for proteomic analysis after circulation is less suitable for human blood sample collection. The first issue is the safety of the material, and even though there are no significant health effects, injecting large amounts of liposomes into the bloodstream may cause problems related to elevated blood lipids. In addition, there may not be sufficient conditions to perform this process when collecting a large number of patient samples in a clinical setting. Therefore, the method can be used as a supplement to existing methods and is more suitable for animal level studies. However, it is also important to consider that the process of the method is cumbersome. The adsorption of proteins based on liposomes may have a certain preference, which can also lead to the loss of information about some proteins. In addition, individual differences in the animal body can lead to a significant difference in liposome uptake and liposome protein adsorption as well, which will make the impact of individual differences in cohort studies even more severe. In fact, in contrast, it may be more feasible to use methods such as DIA proteomics. It can both increase the coverage and quantitative accuracy of protein testing and is a non-invasive method. In a recent study, DIA proteomics has been able to detect on average more than 1300 proteins in plasma (J. Proteome Res. 2024, 23, 9, 3806-3822).

We would like to thank the reviewer for this comment and clarify the following points:

1. Safety and feasibility for human biomarker discovery studies

For the analysis of human clinical plasma samples, NPs are incubated **ex vivo** in a low binding Eppendorf tube with plasma samples obtained from human patients. Therefore, NPs are **not** intravenously injected into human patients, mitigating any risks associated with direct administration. This workflow ensures that the process remains **non-invasive** and suitable for clinical plasma sample analysis.[1]

In our preclinical studies, we chose to intravenously inject NPs into tumour-bearing mice because, as we have previously demonstrated, the *in vivo* protein corona is molecularly more enriched than the *ex vivo* protein corona.[2] *This distinction is explained in our manuscript (Pages 10, Lines 344–347; Fig. 1a and Fig. 5a).*

2. Nano-omics and DIA proteomics

We appreciate the reviewer's comments regarding DIA proteomics, and we agree that it has the potential to increase the number of identified proteins in plasma samples. However, DIA is a data analysis method and cannot replace the need for enrichment or pre-fractionation of plasma samples prior to mass spectrometry analysis.

The Nano-Omics approach functions as a pre-fractionation strategy, specifically developed to remove the overwhelming signal from albumin and other highly abundant proteins in plasma. By

enriching low-abundance proteins, Nano-Omics provides an essential step before mass spectrometry analysis. The reference cited by the reviewer (*J. Proteome Res.* 2024, 23, 9, 3806–3822) also employs a pre-fractionation protocol before applying DIA, underscoring the necessity of robust sample preparation for achieving comprehensive proteome coverage.

While protein corona formation is driven by the affinity of specific plasma proteins for the nanoparticle surface, we and others have previously shown that analysis of protein corona increases the number of identified proteins in comparison to plasma control analysis. The interaction of NPs with plasma proteins is highly influenced by underlying disease pathologies, making it a powerful tool for biomarker discovery.

Future work will aim to combine Nano-Omics with DIA proteomics to investigate whether the detection of low-abundance proteins could be further improved. *We have revised the manuscript to highlight the potential complementary nature of Nano-Omics and DIA proteomics (Page 12, Lines 434-437).*

2. Regarding the innovative nature of this study: The authors claim that there are no reports of nanoparticle-enabled integrative blood-tissue proteomics, which underscores the innovative nature of this approach. There is nothing wrong with this statement. From this perspective, this study is somewhat innovative. However, the authors claim that “To our knowledge, this is the first study in the context of biomarker discovery for GB that attempts to integrate discovery proteomics at the plasma and brain tumour tissue levels” at the end of the manuscript. This statement is not correct. The reviewer note that studies have recently reported molecular mechanism studies and biomarker discovery for gliomas using the integration of blood proteomics and tissue proteomics, and that the entire study was conducted using human samples, which may have more clinical applications than the results based on animal experiments in this study (e.g., *Sci. Adv.* 2024, 10, eadk1721). In addition, this study used laser cutting to obtain both healthy and tumour tissue. While this allows for more precise sampling, it is not innovative, nor is it entirely necessary, except for proteomic analyses at the single-cell level.

We would like to thank the reviewer for bringing the referenced study (*Sci. Adv.* 2024, 10, eadk1721) to our attention. *We have now revised our original statement to reference the study and emphasize that the novelty of our work lies in integration of nanoparticle-enabled plasma proteomics with tumour tissue proteomics (Page 4, Lines 122-127; Page 10, Lines 354-358).*

Regarding the use of laser microdissection for tissue sampling, we agree that this method is not novel. However, we employed it to ensure precise separation of tumour and peri-tumoral regions.

3. The authors observed that the amount of bound protein enriched from the blood of mice became progressively greater as the tumour grew and was attributed to an increase in the material released into the blood stream as the tumour grew. A direct experiment should be added here to verify that the total protein concentration in the blood increases with tumour growth, rather than just speculation. If the total protein concentration does not change significantly, is it possible that the amount of certain proteins is increasing and the liposomes happen to adsorb those proteins preferentially. This is a question worth investigating, since the amount of protein binding is not just slightly increased but almost doubled.

We would like to thank the reviewer for this recommendation. *In response, we have now provided additional data comparing the total amount of protein quantified by BCA assay in plasma samples obtained from GB-bearing mice at days 7, 14, and 18 post-tumour implantations. As shown in **Fig.1** below (now included in the revised manuscript as **Supplementary Figure 1b**), the total amount of plasma proteins remains the same with tumour growth. These results further reinforce the finding that the nanoparticle protein corona enriches GB-specific proteins and/or extracellular vesicles, as also suggested by the increased presence of intracellular proteins with tumour progression (**Supplementary Figure 3b**), indicating greater*

shedding of tumour-released materials or cell debris into the bloodstream. We now better explain this observation in the revised manuscript (Page 5, Lines 178-185).

Fig. 1: Quantification of plasma protein concentration in GL261 tumour-bearing mice. Bar chart displays the plasma protein concentration of tumour-bearing mice at D7, D14 and D18 post-tumour inoculation. Blood recovery was performed using cardiac puncture and the plasma protein concentration was determined using the BCA assay. Error bars represent the mean \pm SD of $n=3-4$ biological replicates. **This analysis has now been added as Supplementary Figure 1b in the revised manuscript.**

4. It is unreasonable that the authors did not use P-value correction for differential protein analysis in the mouse experiment. Although this could have shown more differential proteins, there could have been a large number of false positive results mixed in, which would have affected the reader's judgment of the experimental results. Sometimes more information is not better, because a large amount of mixed information will instead affect our understanding of the truth. Although t-test is a commonly used statistical method in differential protein expression detection to evaluate whether a protein is differentially expressed in two samples by combining variable data between samples. However, the test efficacy of the t-test is reduced due to the small sample size in this study and thus the less accurate estimation of the overall variance, and the number of false positives is significantly increased if the t-test is used multiple times. For example, when the p-value for a protein is less than 0.05 (5%), we usually assume that the expression of this protein is different in the two samples. But there is still a 5% probability that this protein is not a differential protein. Then we incorrectly deny the original hypothesis (no differential expression in the two samples), resulting in a false positive (5% probability of making a mistake). If the test is performed once, the probability of making a mistake is 5%; if it is performed 10,000 times, the number of mistakes made is 500, i.e., an additional 500 additional discrepancies are concluded (even if there are actually no discrepancies). To control for the number of false positives, we need to correct the p-value for multiple comparisons and raise the threshold. Moreover, for label-free methods, proteins with an absolute value of the multiplicity of differences >2 are generally selected for subsequent analysis.

We agree with the reviewer that, for biomarker discovery, the *FDR adjusted p-value (q-value)* should be used to identify differentially abundant proteins (DAPs) with potential as biomarkers. In our human pilot study, **all potential biomarkers shortlisted in Fig. 5 represent proteins with a q-value < 0.05** , identified in both human blood and tumour tissues. These proteins are also associated with the seven enriched pathways observed in the mouse data analysis. Therefore, we believe this final list of human proteins constitutes a robust set of potential biomarkers for further validation studies.

In the mouse studies, our primary goal was not to identify potential biomarker proteins but rather to correlate changes in the plasma proteome with those occurring in brain tissue. This approach was aimed at enhancing our understanding of the underlying molecular mechanisms. Consequently, in discussing the mouse data, we do not claim to have discovered potential biomarkers. Instead, we focus on identifying molecular pathways that could potentially serve as sources for peripheral blood biomarkers.

Given that only three biological replicates were performed for each time point in the mouse experiments, we believe that including only proteins with an *adjusted p-value* < 0.05 would

significantly compromise our mechanistic understanding of disease progression, leading to a higher rate of false negatives. This is why less stringent thresholds are typically employed in preclinical exploratory studies compared to human studies, which often include larger numbers of biological replicates. For these reasons, we chose not to exclude any DAPs from our analysis.

We agree with the reviewer however, that this could potentially affect the reader's judgment of the experimental results. We have therefore, made the following revisions: a) Added the adjusted p-values (q-values) to all Supporting Tables (including the mouse experiments); b) Included fold change > 2 threshold lines in all Volcano Plots (mouse and human experiments); c) Revised all figure legends to clearly state the DAPs selection criteria; and d) Expanded the explanation of the rationale behind this mouse study analysis and revised any misleading statements with regards to number of DAPs or the identification of potential biomarker proteins in the mouse work (Page 5, Lines 189-196).

5. In the analysis of tissue samples, equivalent volumes of brain tissue from sham-injected control mice were also subjected to LC-MS/MS analysis to account for any changes in the brain tissue proteome caused by the intracranial injection. But the results of the analysis and discussion of this data are not seen in the manuscript.

We would like to thank the reviewer for this recommendation. *We have now added volcano plots of DAPs identified between sham-injected hemisphere and contralateral (non-injected) hemisphere for the three different time points (Supplementary figure 3c and Supplementary Tables 7-9). The proteins plotted in the volcano plots below were removed from the list of DAPs reported in Figure 4b. This is now mentioned in the revised manuscript (Page 7, Lines 264-267).*

Fig. 2: Volcano plots showing the relationship between fold change and significance for DAPs identified through comparisons between sham-injected hemisphere and contralateral (non-injected) hemisphere from sham-injected control mice for the three different time points. **This analysis has now been added as Supplementary Figure 3c, in the revised manuscript.**

Reviewer #3 (Remarks to the Author):

Xinming Liu and colleagues, have revised the manuscript and accurately answered point by point to all the comments. Despite I acknowledge the effort in discussing some of the raised criticisms, the most important aspect about the specificity of the biomarkers composing the protein corona for GB patients has not been clarified. Although we all agree that a panel of biomarkers is better than a single marker and that we may witness the formation of "personalized corona", the protein composition of the corona should have a clear biological significance and specificity with respect to the disease studied to be useful in clinic. The proteins mentioned remain clearly non-specific for glioblastoma. They are proteins already discovered in the past, associated to other cancers and disease.

We appreciate the reviewer's acknowledgment of our efforts to address the comments and the opportunity to further clarify the disease specificity of the protein corona and the clinical utility of the potential biomarkers identified. *Below, we outline the additional steps taken to address these concerns:*

1. Specificity of Protein Corona Biomarkers

We have now performed comprehensive literature search of the 48 shortlisted biomarker candidates which were identified in our study through the analysis of both human plasma and tumour tissue and were additionally found to be involved in the seven pathways commonly identified in both mice and humans. As shown in the table below (*now added as **Supplementary Figure 6** in the revised manuscript*):

- Out of the 48 proteins, 11 proteins have previously been reported in the plasma of GBM patients, the majority of which have also been reported in the plasma of other cancers.
- We also report the discovery of **16 proteins (highlighted in blue)** that have not been previously reported at the plasma level in glioblastoma or any other cancer type, suggesting the identification of newly discovered plasma biomarkers for glioblastoma. We have now discussed these findings in Page 12, Lines 428-434 of the revised manuscript.

Human plasma-to-tumour tissue shared proteins					
Gene name	Description	Plasma p value	Plasma fold change	Previously reported in GBM plasma	Previously reported in plasma of other cancers
ACTB	Actin, cytoplasmic 1	3.67E-05	4.31	No	Yes
ACTR3	ARP3 actin-related protein 3 homolog	6.80E-05	6.36	No	No
ARPC1B	Actin-related protein 2/3 complex subunit 1B	2.80E-05	11.85	No	No
ARPC2	Actin-related protein 2/3 complex subunit 2	1.65E-05	8.76	No	Yes
ARPC4	Actin-related protein 2/3 complex subunit 4	1.85E-04	30.36	No	No
CDC42	Cell division control protein 42 homolog	3.07E-04	2.55	No	Yes
COL6A2	Collagen alpha-2(V) chain	1.22E-02	-1.53	No	No
CORO1A	Coronin	2.37E-05	8.74	No	Yes
CYFIP1	Cytoplasmic FMR1-interacting protein 1	2.37E-05	7.93	No	No
ENO1	Enolase 1 (Alpha)	1.23E-04	3.59	No	Yes
F2	Fibrinogen	4.68E-03	3.27	Yes	Yes
FCER1G	High affinity immunoglobulin epsilon receptor subunit gamma	3.97E-05	3.26	No	No
FGA	Fibrinogen alpha chain	2.95E-02	1.88	Yes	Yes
FN1	Fibronectin 1	1.15E-03	-2.26	No	Yes
GAPDH	Glyceraldehyde-3-phosphate dehydrogenase	1.24E-04	3.6	No	Yes
GNAI2	Guanine nucleotide-binding protein G(i) subunit alpha-2	2.20E-02	1.71	No	Yes
ILK	Integrin-linked protein kinase	2.70E-06	7.79	No	Yes
ITGB1	Integrin beta-1	6.27E-04	3.03	Yes	Yes
MYH9	Myosin heavy polypeptide 9	1.97E-05	4.19	Yes	Yes
MYL12A	Myosin regulatory light chain 12A	1.17E-03	12.78	No	No
NCKAP1	Nck-associated protein 1	1.22E-02	11.91	No	No
PFKL	ATP-dependent 6-phosphofructokinase	8.44E-03	Infinity	No	No
PGK1	Phosphoglycerate kinase 1	1.31E-05	8.24	Yes	Yes
RAC2	Ras-related C3 botulinum toxin substrate 2	3.48E-05	3.4	No	No
TLN1	Talin-1	3.10E-06	14.98	No	Yes
TP11	Triosephosphate isomerase	1.25E-02	3.62	Yes	Yes
TUBA4A	Tubulin alpha-4A chain	8.06E-07	35.37	No	Yes
TUBB	Tubulin beta chain	3.80E-05	5.32	No	Yes
VASP	Vasodilator-stimulated phosphoprotein isoform 1	1.04E-03	6.99	No	Yes
WWF	von Willebrand factor	2.90E-07	29.94	Yes	Yes
WASF2	Wiskott-Aldrich syndrome protein family member 2	1.45E-05	7.57	No	No
Human plasma-to-tumour tissue shared proteins (identified in mouse plasma)					
Gene name	Description	Plasma p value	Plasma fold change	Previously reported in GBM plasma	Previously reported in plasma of other cancers
ALDOA	Fructose-bisphosphate aldolase A	3.49E-05	8.06	Yes	Yes
ARPC5	Actin-related protein 2/3 complex subunit 5	1.38E-04	58.68	No	No
C5	Complement C5	6.78E-03	1.95	Yes	Yes
C8A	Complement component C8 alpha chain	4.62E-02	-1.97	No	Yes
ITGA2	Integrin alpha-2	1.61E-02	3.48	No	Yes
LYN	Tyrosine-protein kinase Lyn	6.47E-05	8.54	No	No
RAC1	Ras-related C3 botulinum toxin substrate 1	2.23E-03	2.7	No	Yes
Human plasma-to-tumour tissue shared proteins (identified in mouse plasma and tumour tissue)					
Gene name	Description	Plasma p value	Plasma fold change	Previously reported in GBM plasma	Previously reported in plasma of other cancers
CFL1	Cofilin-1	3.94E-03	4.27	No	No
FGB	Fibrinogen beta chain	3.34E-02	1.79	No	Yes
FGG	Fibrinogen gamma chain	8.02E-03	1.73	No	Yes
GNAI3	Guanine nucleotide-binding protein G(i) subunit alpha	2.14E-03	3.08	No	No
ITGA6	Integrin alpha-6	1.05E-06	11.98	No	Yes
PFN1	Profilin-1	6.42E-04	5.69	No	Yes
RAP1B	RAP1B, member of RAS oncogene family	8.18E-05	6.14	Yes	Yes
TNC	Tenascin C	2.54E-02	1.76	Yes	Yes
TUBB4B	Tubulin beta-4B chain	2.49E-06	9.42	No	No
VCL	Vinculin	9.67E-05	9.29	No	Yes

Table 1: Full list of the n=48 human plasma-to-tumour tissue shared proteins involved in 7 pathways found to be shared between both human and mouse species: the regulation of actin cytoskeleton, focal adhesion, platelet activation, leukocyte transendothelial migration, biosynthesis of amino acids, carbon metabolism and phagosome. Out of the 48 human DAPs mapped to the seven pathways, 17 were also found to exhibit differential abundance in the plasma of GL261 tumour-bearing mice, with 10 of these also identified in the mouse tumour tissue. Proteins not previously reported at the plasma level in GB, or any other cancer type are highlighted in blue. This table has now been added in Supplementary Figure 6 of the revised manuscript.

2. *Clinical Utility of Plasma Biomarkers in the context of Glioblastoma*

While we acknowledge that molecular mechanisms and proteins may overlap between different cancer types, it is important to consider the need for blood biomarkers in the clinical context of glioblastoma. Patients with glioblastoma often present with non-specific symptoms (e.g. headache), and a blood test would primarily serve to identify individuals suspected of having glioblastoma who should then undergo follow-up diagnostic imaging, ideally within a two-week window, using MRI. It is worth noting that no existing cancer blood test is intended to stand alone as a definitive diagnostic tool without confirmation from other diagnostic modalities.

In our study, the protein corona analysis plays a critical role in analyzing the plasma proteome and identifying a list of potential biomarker proteins that are commonly identified in our tumour tissue analysis. These proteins will be validated in subsequent studies using larger cohort clinical samples by targeted immunoapproaches, *directly applied to plasma samples without the need for further incubation with nanoparticles*. Therefore, protein corona formation is solely used as a tool to address the issue of albumin masking and enable deeper analysis of the plasma proteome (similar to how plasma fractionation steps work). By comparing protein corona in healthy and diseased individuals, our aim is to identify DAPs and by then intergating the data with tumour tissue proteomics we hope to shortlist the identified DAPs for further validation. *This is now comprehensively explained in the revised manuscript (Page 12, Lines 440-453).*

The study remains interesting from a technical point of view, because it allows studying the formation and composition of the crown in relation to the size of the tumour (even if tumour size at the different times is not specified). But it cannot be published with the title "*Nano-omics Enables Plasma-to-Tumour Tissue Integrated Proteomics for Biomarker Discovery in Glioblastoma*", because it does not in fact demonstrate the discovery of any new biomarker or group of markers contained in the corona specific for the GB. Interesting would be at least to see a comparison between the candidate protein biomarkers identified in this study and those identified with a similar methodology in other cancer types (even if just at a literature level), proving some potential specificity.

We thank the reviewer for for finding the technical aspects of our study interesting. We would like to clarify that tumour size at the different time points is indeed provided in the manuscript and can be found in **Fig.1c**.

We also hope that the table presented in **Supplementary Figure 6** demonstrates the discovery of a new group of potential biomarker proteins for glioblastoma. We would respectfully like to retain the current title of our manuscript, as we believe it accurately reflects the scope of our work. In biomarker discovery studies, it is standard practice to compare plasma samples from patients to those from healthy control individuals in order to identify potential biomarkers. The inclusion of additional control groups from other cancer types is typically part of **validation studies**, which do not necessarily require discovery proteomics but rely on ELISA validation using targeted immunoassays.

In response to the reviewer's suggestion, *we have now also compared the candidate protein biomarkers identified in our study with proteomics data we have previously obtained from an ovarian cancer cohort* [1]. It should be noted that the same *ex vivo* pipeline was applied to plasma samples from both ovarian cancer and glioblastoma patients, ensuring comparability between the datasets. For this comparative analysis, we included DAPs with an adjusted *p-value* < 0.05 and that were associated with a gene name. As shown in the Venn diagram of **Fig.3** below, a total of 118 potential biomarkers were common between the two cancer types; however, the majority of DAPs were unique to each cancer cohort (154 DAPs for the ovarian cancer cohort; 297 DAPs for the glioblastoma cohort). This finding suggests that the nanoparticle protein corona consists of both cancer-associated proteins and cancer-type-specific proteins.

Fig. 3. Venn diagram compares the number of common and unique plasma DAPs identified in glioblastoma and ovarian cancer patients by the analysis of the ex vivo protein corona. Only proteins with an adjusted-p value <0.05 and associated with a gene name were included in the analysis.

We greatly appreciate the reviewer's thoughtful comment regarding the specificity of the protein corona. We fully recognize the importance of understanding the disease-specific nature of the protein corona and have analysed these data in response to the reviewer's comment. However, we have decided not to include these findings in the current manuscript, as they form the basis of a more comprehensive, in-depth subsequent publication that will specifically address the disease specificity of the protein corona. The primary aim of this manuscript is the nanotechnology-enabled integration of blood and brain tissue proteomics in glioblastoma, and we wish to maintain a clear and focused narrative of our study. *We now better discuss the disease specificity of the protein corona and the need for further validation studies in GB, in our revised manuscript (Page 10, Lines 347-358; Page 12, Lines 440-453).*

Reviewer #4 (Remarks to the Author):

The manuscript presents an innovative approach utilizing in vivo protein corona analysis to monitor protein variations during the progression of glioblastoma in a mouse model. By comparing these findings with the protein corona composition of human plasma samples, the study identifies overlapping proteins that hold potential clinical value for the early detection of this aggressive disease. The integration of animal model data with human samples offers a promising pathway toward translational applications in glioblastoma diagnostics. I have thoroughly reviewed the manuscript, including the reviewer comments and the authors' responses to the raised concerns. The additional experiments and explanations provided by the authors effectively address the initial critiques, enhancing the study's robustness and credibility. I am inclined to recommend publication of this insightful paper, pending the incorporation of the following revisions:

We would like to thank the reviewer for the thorough evaluation of our manuscript and their supportive remarks regarding the innovative approach and translational potential of our study. We are pleased that the additional experiments and explanations have successfully addressed the initial reviewer comments. *We appreciate the recommendation for publication and have carefully incorporated the suggestions provided to further strengthen the manuscript.*

1. The manuscript should provide a more detailed description of the mass spectrometry techniques used. Specifically, it is important to clarify whether all raw protein corona data were compiled and subjected to a harmonized analysis. This approach has recently been shown to significantly improve the robustness and accuracy of the protein corona outcomes.

We agree with the reviewer that, for mass spectrometry analysis, the harmonization of sample digestion protocols, instrument settings, and data acquisition parameters is essential to enhance the reproducibility of the results. Accordingly, we have ensured that all raw protein corona data were compiled and subjected to a harmonized analysis pipeline. Furthermore, to minimize variability, all samples obtained from the mouse and human experiments were processed and run sequentially. All raw protein corona data were compiled and subjected to a harmonized analysis, aligning with recent recommendations in nanoparticle corona analysis [3]. *This is now better explained in the methods section of the revised manuscript (Page 22, Lines 785-786 and 799-800).*

2. The authors should expand their discussion to consider the implications of personalized

protein corona, taking into account factors such as sex, age, and other individual-specific variables. Understanding how these factors influence protein corona composition is essential for assessing the generalizability and applicability of the findings.

We would like to thank the reviewer for this recommendation. We agree that factors such as sex, age, and other individual-specific variables can play a significant role in shaping the protein corona composition [4]. *We have now expanded the discussion about the implications of personalized corona in the revised manuscript (Page 10, Lines 346-357).*

3. It is important to delineate whether the identified protein patterns are specifically associated with glioblastoma or if they are influenced by individual biological differences. This distinction is critical for validating the identified proteins as reliable biomarkers for glioblastoma. A practical approach would be to conduct protein corona analyses on plasma samples from both male and female subjects and examine the patterns of key proteins. Additionally, stratifying samples based on age groups or other relevant demographics could help isolate disease-specific protein signatures from those influenced by individual variability. Incorporating these analyses would not only enhance the study's depth but also provide a more comprehensive understanding of how personalized factors may affect protein corona composition. This, in turn, would refine the proposed proteins as potential biomarkers, thereby increasing their clinical relevance and utility in glioblastoma detection.

We agree with the reviewer that individual biological differences can significantly influence the composition of the plasma proteome and, consequently, the formation of protein corona.

To address this biomarker specificity challenge, our study integrates plasma proteomics with glioblastoma tumour tissue proteomics. To minimize variability between patients, we employed the following steps:

1. Sex- and Age- Matched Controls

Our study includes both male and female patients, and sex- and age-matched healthy controls were used (**Supplementary Figure 4a**). However, we acknowledge that the small sample size in our human cohort experiment (n=6 males; n=4 females) limits our ability to draw definitive conclusions regarding the sex-specificity of the candidate biomarkers identified. Consequently, the clinical utility of the aforementioned DAPs as early detection and/or disease-monitoring biomarkers for glioblastoma will require future validation in larger, longitudinal patient cohorts that include a diverse group of patient and control cohorts. These future studies will necessitate the use of targeted immunoapproaches to quantify the potential protein biomarkers in plasma, which is beyond the scope of the current study. *This is now better discussed in the revised manuscript (Page 12, Lines 440-453).*

2. Multi-Dimensional Integrative Proteomics Workflow

As illustrated in **Supplementary Figure 5a**, our workflow ensures the selection of glioblastoma-specific biomarkers by systematically filtering the identified 321 DAPs using the following steps:

- Highlighting 140 proteins also found in surgically resected tumour tissues.
- Narrowing these to 61 proteins identified in our longitudinal mouse study to ensure cross-species relevance.
- Identifying seven pathways commonly observed in both human and mouse datasets.
- Further narrowing the human DAPs to 48 proteins specifically involved in these seven pathways.

*We have now conducted a comprehensive literature review of the 48 shortlisted biomarker candidates. A summary of this review has now been included in the revised manuscript as **Supplementary Figure 6**. Out of the 48 proteins, 11 proteins have previously been reported in*

the plasma of GBM patients, the majority of which have also been reported in the plasma of other cancers. We also report the discovery of 16 proteins (that have not been previously reported at the plasma level in glioblastoma or any other cancer type, suggesting the identification of newly discovered plasma biomarkers for glioblastoma (Page 12, Lines 428-437).

References:

- 1) Hadjidemetriou M, Papafilippou L, Unwin R, Clamp A, Kostarelos K. Nanoparticle-enabled cancer biomarker discovery: a proof of concept study in ovarian carcinoma patients. *Nano Today*, **2020**, 34, 100901; DOI: 10.1016/j.nantod.2020.100901.
- 2) Hadjidemetriou M, Al-Ahmady Z, Mazza M, Collins RF, Dawson K, Kostarelos K. In vivo biomolecule corona around blood-circulating, clinically used and antibody-targeted lipid bilayer nanoscale vesicles. *ACS Nano*, **2015**, 9, 8142-56; DOI: 10.1021/acs.nano.5b03300.
- 3) Gharibi, H., Ashkarran, A.A., Jafari, M. et al. A uniform data processing pipeline enables harmonized nanoparticle protein corona analysis across proteomics core facilities. *Nat Commun*, **2024**, 15, 342; <https://doi.org/10.1038/s41467-023-44678-x>.
- 4) Mahmoudi, M., Landry, M.P., Moore, A. et al. The protein corona from nanomedicine to environmental science. *Nat Rev Mater*, **2023** 8, 422–438; <https://doi.org/10.1038/s41578-023-00552-2>.